# Dissection and function of autoimmunity-associated *TNFAIP3* (A20) gene enhancers in humanized mouse models

Upneet K. Sokhi [1], Mark P. Liber[1], Laura Frye[1], Sungho Park[1], Kyuho Kang [1], Tania Pannellini[2], Baohong Zhao[1], Rada Norinsky[3], Lionel B. Ivashkiv [1,4,5] & Shiaoching Gong[1,3]

Enhancers regulate gene expression and have been linked with disease pathogenesis. Little is known about enhancers that regulate human disease-associated genes in primary cells relevant for pathogenesis. Here we use BAC transgenics and genome editing to dissect, in vivo and in primary immune cells, enhancers that regulate human *TNFAIP3*, which encodes A20 and is linked with autoimmune diseases. A20 expression is dependent on a topologically associating subdomain (sub-TAD) that harbors four enhancers, while another >20 enhancers in the A20 locus are redundant. This sub-TAD contains cell- and activation-specific enhancers, including an enhancer (termed TT>A) harboring a proposed causal SLE-associated SNV. Deletion of the sub-TAD or the TT>A enhancer results in enhanced inflammatory responses, autoantibody production, and inflammatory arthritis, thus establishing functional importance in vivo and linking enhancers with a specific disease phenotype. These findings provide insights into enhancers that regulate human A20 expression to prevent inflammatory pathology and autoimmunity.

---

[1] Arthritis and Tissue Degeneration Program, David Z. Rosensweig Center for Genomic Research, Hospital for Special Surgery, New York, NY 10021, USA. [2] Research Division and Department of Pathology, Hospital for Special Surgery, New York, NY 10021, USA. [3] Rockefeller University, New York, NY 10065, USA. [4] Graduate Program in Immunology and Microbial Pathogenesis, Weill Cornell Medicine, New York, NY 10065, USA. [5] Department of Medicine, Weill Cornell Medicine, New York, NY 10065, USA. Upneet K. Sokhi and Mark P. Liber contributed equally to this work. Lionel B. Ivashkiv and Shiaoching Gong jointly supervised this work. Correspondence and requests for materials should be addressed to L.B.I. (email: ivashkivl@hss.edu) or to S.G. (email: gongs@rockefeller.edu)

Multiple enhancers have been identified on the basis of chromatin states and accessibility, transcription factor binding, sequence conservation, and have been linked to target genes/promoters based upon proximity or chromatin conformation (looping)[1]. Genes are typically associated with several enhancers, but little is known about which enhancers are functional, the importance of individual enhancers, and whether they function in an additive or context- or signal-dependent manner. Indeed, a conundrum in the field is that many putative enhancers identified by epigenomic approaches have not yet been shown to be functionally important in regulating gene expression. Studies in model organisms have identified functional enhancers, often enhancers important for expression of developmental genes, but little is known about enhancer function in mammalian primary cells[1]. Recent studies have begun to address mammalian enhancer function using CRISPR-mediated genome editing, including in vivo analysis of mouse genes[2–5]. In vivo dissection of small superenhancers (containing 3–5 individual enhancers) associated with the mouse α-globin, Wap, and Nr4a1 genes revealed that individual enhancers are mostly independent of each other and function in an additive manner, although there is a hierarchy of relative enhancer strength[2,3,5]. Analysis of human gene enhancers has been limited by use of artificial systems that do not capture chromatin context, or are performed in cell lines that may not reflect regulation in primary cells and are not relevant for physiology or disease pathogenesis. As regulatory elements such as enhancers are not well conserved[6], understanding regulation of human genes requires studying human enhancers in primary cells. Such studies are highly relevant for understanding disease pathogenesis, as disease-associated allelic variants (single-nucleotide variants; SNVs) are enriched in enhancers[7,8], and identifying important functional enhancers can help identify causal SNVs and yield insight into mechanisms of disease.

The A20 protein encoded by TNFAIP3 is a key negative regulator of nuclear factor-κB (NF-κB)-mediated inflammatory signaling. Deficiency of A20 in immune cells leads to systemic inflammation and autoimmunity[9–11]. Multiple SNVs in TNFAIP3 that encodes A20 have been tightly linked with various autoimmune diseases. Disease-associated TNFAIP3 SNVs fall mostly in noncoding regions, and likely confer disease susceptibility by affecting the function of enhancers[12–14]. Interestingly, reduced A20 expression has been documented in cells from patients with several autoimmune diseases, and reduced A20 expression and polymorphisms in TNFAIP3 have been associated with response to tumor necrosis factor (TNF)-blocking therapies in rheumatoid arthritis (RA), psoriasis, and Crohn's disease[9,15]. We wished to understand the regulation of human TNFAIP3 and identify functionally important enhancers that are relevant for autoimmune diseases in vivo. Bacterial artificial chromosome (BAC)-humanized mouse models expressing disease-associated genes in their genomic context offer the potential to dissect the function of human autoimmunity-associated enhancers in vivo and in autoimmune disease models. We took the approach of transgenic expression in mice of TNFAIP3-containing human BACs, large-insert genomic DNA fragments that are typically insulated from position effects and are faithfully expressed in transgenic mice[16–19]. We first generated three humanized mouse lines to survey large genomic regions flanking the human TNFAIP3 gene for cis-regulatory function. All three BACs rescue the perinatal lethal phenotype of Tnfaip3 global KO mice, but only deletion of downstream sequences compromised human A20 (hA20) expression, in lymphocytes, macrophages, and synovial fibroblasts (SFs). Further application of CRISPR- and recombineering-mediated genome editing on downstream enhancer regions reveals a key role for an enhancer harboring a TT>A disease-

associated SNV in A20 expression. Lower hA20 expression in mice harboring BACs missing downstream regulatory sequences or the TT>A enhancer resulted in growth retardation, increased susceptibility to endotoxin toxicity, spontaneous autoantibody production, and inflammatory arthritis. These results map genomic regions important for hA20 expression that may have relevance for control of immune responses and autoimmunity, and set the stage for fine mapping and identification of functional enhancers.

## Results

**Epigenomic analysis of TNFAIP3 enhancers and BAC selection.** To guide selection of BACs for generation of humanized transgenic mice, we analyzed chromatin conformation, accessibility, and enhancer-related histone marks at the TNFAIP3 locus using our laboratory's (GSE43036[20], GSE100383[21] and previously unpublished data, GSE104628) and ENCODE and NIH Roadmap ChIP-seq data (available at genome.ucsc.edu and road-mapepigenomics.org) and Hi-C data (GSE63525[22]). Analysis of pre-existing high-resolution Hi-C (chromatin conformation) data in GM12878 B cells identified a 305 kb topologically associating domain (TAD) flanked by CTCF sites roughly centered around the TNFAIP3 gene body (Fig. 1a, central black box and Supplementary Fig. 1a, bottom track). The TNFAIP3-containing TAD did not harbor any additional genes and was comprised of three subdomains (sub-TADs; enclosed in yellow boxes, Fig. 1a), two upstream and one downstream of the TNFAIP3 transcription start site (TSS) and gene body (TSS marked by arrow, Fig. 1a). The strongest looping interactions (mediated by anchors, shown as blue boxes in Fig. 1a) were between the promoter and the distal portions of the sub-TADs immediately upstream and down-stream of the TSS (Fig. 1a, top). Similar looping interactions were detected in a distinct data set[23] and are visualized in Supplementary Fig. 1b. This analysis defines regions predicted to harbor enhancers that regulate TNFAIP3.

To define putative enhancers within the TNFAIP3 TAD, we used the standard approach of identifying regions of open chromatin that are marked by histone 3 lysine 4 monomethyla-tion (H3K4me1)[1]. Analysis of pre-existing ENCODE and NIH Roadmap data for monocytes, B cells, T cells, and IMR90 fibroblastic cells identified >30 potential enhancers within the TNFAIP3 TAD (Supplementary Fig. 1a). The high density of enhancers is indicative of a super-enhancer, which is typically contained together with its target gene within a TAD[24,25]. The TNFAIP3 locus contains putative enhancers common to all cell types analyzed as well as cell type-specific enhancers. Autoimmune disease-associated SNVs occur throughout this locus and overlap with or are in linkage disequilibrium with various enhancers (Supplementary Fig. 1a, top). TNFAIP3 expression is induced by inflammatory stimuli such as LPS and TNF. To identify enhancers activated by inflammatory stimuli, we analyzed ATAC-seq and ChIP-seq data sets generated in our laboratory (GSE43036, GSE98369[26], and GSE104638) for chromatin accessibility, binding of PU.1 and C/EBP (which bind to regulatory elements and help define enhancers in myeloid cells), and induction of acetylation of histone 3 lysine 27 (H3K27-Ac), which indicates enhancer activation, at enhancer peaks after LPS stimulation of primary monocytes (Fig. 1b). The TNFAIP3 locus exhibited multiple enhancers in resting monocytes; consistent with basal expression of A20, a fraction of monocyte enhancers exhibited H3K27-Ac at baseline. Stimulation with LPS-induced C/EBP binding at a subset of enhancers and massive and broad H3K27-Ac at enhancers across the entire locus (Fig. 1b). This epigenomic data suggested multiple enhancers at the TNFAIP3 locus could be functionally important in basal and stimulus-

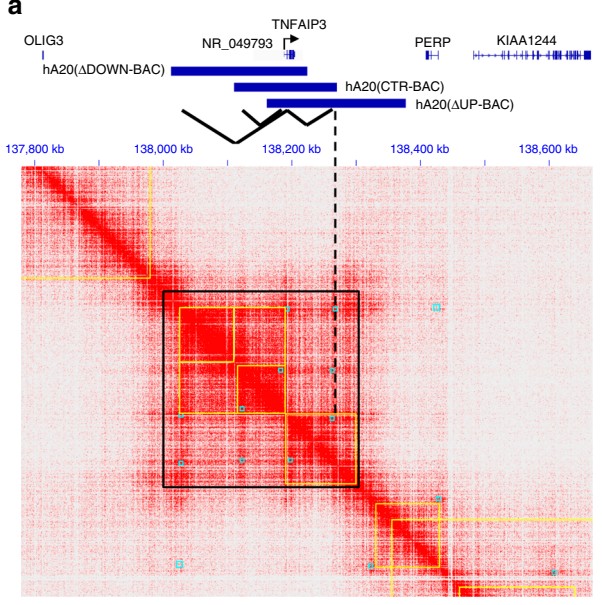

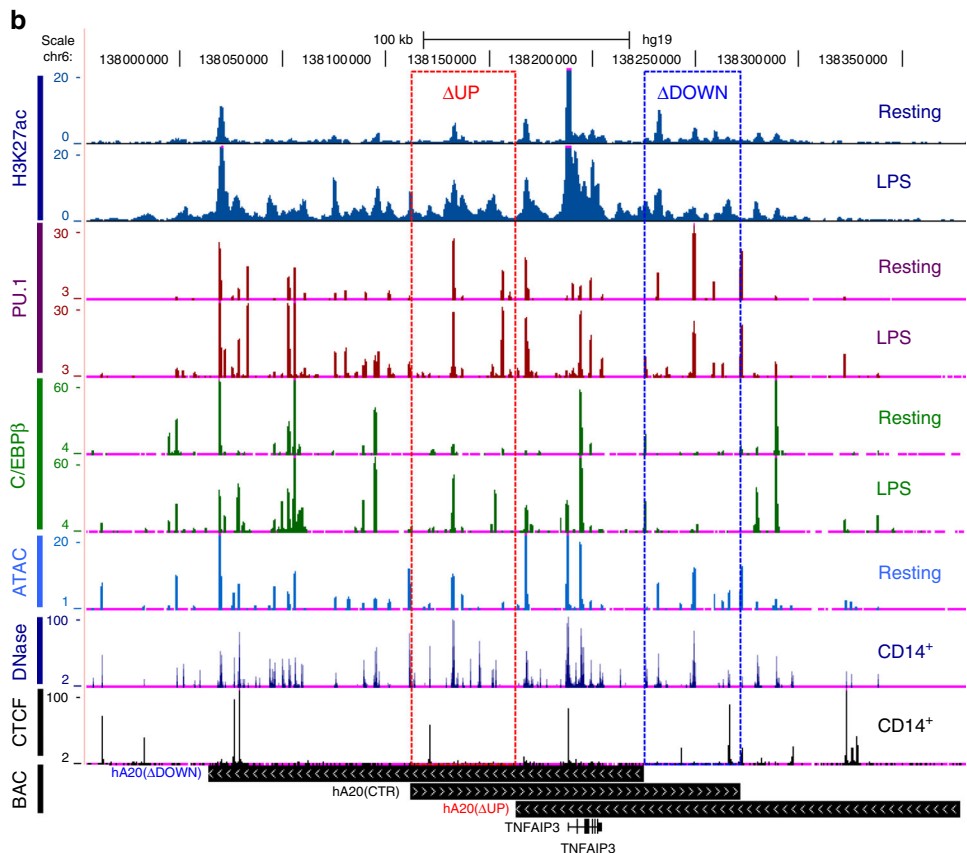

**Fig. 1** Epigenomic analysis of human TNFAIP3/A20 locus. **a** Heat map showing genomic region interactions based on Hi-C data in GM12878 B cells. Yellow boxes indicate topologically associating (sub) domains and blue boxes, interaction anchors. Dashed line shows downstream anchor relative to BAC locations. **b** Gene track showing chromatin accessibility (ATAC-seq, DNase) or H3K27-Ac, PU.1, C/EBP, and CTCF ChIP-seq read counts in primary human monocytes stimulated −/+LPS for 3 h. Genomic locations of BACs used in this study are shown. Data are representative of two replicates or derived from public databases

responsive *TNFAIP3* expression. The existence of multiple enhancers supported a strategy of initially deleting large genomic regions to identify regions that harbor the most important enhancers. Working within the constraints of upper limits of BAC size of 150–200 kb, we selected three BACs for initial

generation of transgenic mice (shown in Fig. 1a upper and b lower sections, and Supplementary Table 1). These BACs essentially serve as deletion mutants relative to each other that can be compared to map functional enhancers, and are termed hA20(CTR-BAC) (centered on the TSS with intact flanking

upstream and downstream sub-TADs), hA20(ΔUP-BAC) (lacking >20 upstream enhancers), and hA20(ΔDOWN-BAC) (lacking 4 enhancers in the downstream sub-TAD). We reasoned that comparison of these BACs would allow us to determine the functions and relative importance of upstream and downstream regulatory regions.

**Functionally important sub-TAD harboring four enhancers**. Multiple *TNFAIP3* BAC transgenic lines were generated and gene copy number-dependent expression confirmed as described in Methods and Supplementary Figs. 2 and 3. For further analysis, we focused on matched mouse lines that harbored one hA20 BAC transgene copy, and confirmed results using additional mouse lines. The mouse lines with one copy of each transgene crossed onto the mA20 null background used in this study are termed hA20(CTR), hA20(Δ-UP), and hA20(Δ-DOWN) to correspond to the BACs described above. To determine if the human BAC transgenes have a similar profile of enhancer peaks in mouse as in human cells, we performed formaldehyde-assisted isolation of regulatory element (FAIRE) experiments in autoimmune disease-relevant cell types—splenic B cells and bone marrow-derived macrophages (BMDMs). The majority of regulatory regions tested in mouse cells exhibited open chromatin (Supplementary Fig. 4a, left panels) that overlapped with the enhancer and promoter regions in human cells identified in Fig. 1. The pattern of open chromatin and responsiveness (increased chromatin accessibility) to cell activation was similar in B cells and in BMDMs, except that chromatin was more open at the TT>A enhancer (see below) in B cells than in monocytes. Deletion of upstream or downstream enhancers had minimal effect on chromatin accessibility at the remaining enhancers (Supplementary Fig. 4b). To study if the BAC transgenes lacking defined regulatory elements exhibit differences in basal and cytokine-inducible expression of hA20, we analyzed hA20 mRNA and protein expression in these cell types. Similar to the pattern of mA20 mRNA expression[27], hA20 mRNA was expressed at baseline in transgenic cells, and was induced when cells were activated using various stimuli (Fig. 2a, b). Basal and inducible expression of hA20 mRNA was mostly similar in hA20(CTR) and hA20(Δ-UP) cells, except that basal hA20 mRNA expression was significantly higher in Δ-UP BMDMs (Fig. 2a), which was also reflected at the protein level (Fig. 2c, middle panels and Supplementary Fig. 15a, b). Basal expression of hA20 was also consistently elevated in SFs. Thus, the upstream region may contain negative regulatory elements that suppress basal *TNFAIP3* expression. In contrast, basal and inducible hA20 mRNA was substantially and significantly reduced in hA20(Δ-DOWN) cells. This reduction in hA20 expression was confirmed using independent mouse lines (Supplementary Fig. 3a, b). In accord with the mRNA results, basal and inducible hA20 protein amounts were lower in B cells, BMDMs, and SFs from hA20 (Δ-DOWN) relative to hA20 (CTR) or hA20 (Δ-UP) transgenic mice (Fig. 2c and Supplementary Figs. 5c, d, 11a; lower band is specific for A20). These results suggest that four enhancers contained in the sub-TAD downstream of the *TNFAIP3* gene body are important for hA20 mRNA and protein expression in several hematopoietic cell types and in mesenchymal lineage fibroblasts.

**Downstream enhancers prevent inflammatory pathology**. Induction of cytokines and chemokines by activating stimuli are tightly regulated by A20 via negative regulation of NF-κB pathway signaling. Important functional consequences of A20 deficiency include increased cytokine production upon cell activation and eventual development of autoimmunity. We observed a significant increase in induction of TNF and a trend toward higher

interleukin (IL)-6 in hA20(Δ-DOWN) cells (Fig. 2d), but overall the effects of diminished hA20 expression on cytokine production in vitro were modest (Supplementary Fig. 6). This is most likely because a near complete deficiency of A20 is required to observe increased cytokine production in the context of strong in vitro activation, as even an ~90% decrease in A20 resulted in small changes in cytokine production in patients with A20 mutations[11]. In contrast, elevated amounts of CXCL10 protein were clearly observed in vivo in the serum of aged (>6 months old) hA20 (Δ-DOWN) mice (Fig. 2e); elevated CXCL10 expression has been observed in autoimmune diseases such as systemic lupus erythematosus (SLE)[28,29].

As regulatory element mutations causing relatively small changes in gene expression can have striking effects in vivo[30], we analyzed the effects of deletion of the downstream sub-TAD on transgenic mouse phenotypes related to autoimmunity. When crossed onto a mouse *Tnfaip3−/−* (mA20-deficient) background, all single-copy BAC transgenes rescued the perinatal lethal phenotype of mA20-deficient mice[9] (Supplementary Table 2). Thus, all three hA20 BAC transgenes are sufficiently expressed to prevent a strong lethality phenotype. However, hA20(Δ-DOWN) mice housed under specific pathogen-free (SPF) conditions exhibited a striking and significant ($P = 0.003$) suppression of their growth curves that was clearly apparent as soon as 4 weeks after weaning and persisted throughout the observation period (mean 15% lower weight in female mice; Fig. 3a). This result is consistent with runting of *Tnfaip3−/−* mice that occurs secondary to inflammation driven by endogenous microbial flora under SPF conditions[9]. Growth retardation was also observed in male transgenic mice and in C57BL/6 mice reconstituted with hA20 (Δ-DOWN) bone marrow (Fig. 3b and Supplementary Fig. 7a). Thus, absence of downstream *TNFAIP3* enhancers in hematopoietic cells is sufficient to confer a growth retardation phenotype; although it is possible that decreased hA20 expression in other cell types can also play a role. These results suggest that the four enhancers in the downstream sub-TAD that are absent in hA20(Δ-DOWN-BAC) are functionally important in hematopoietic cells in vivo.

To obtain additional evidence that decreased hA20 expression in hematopoietic cells, as observed in hA20(Δ-DOWN) chimeric mice, could contribute to increased inflammatory responses in vivo, we subjected C57BL/6 mice reconstituted with hA20 (CTR), hA20(Δ-UP), or hA20(Δ-DOWN) bone marrow to LPS challenge. In this model, LPS injection induces a cytokine storm that leads to toxicity and lethality. Strikingly, mice reconstituted with hA20(Δ-DOWN) bone marrow showed early and increased lethality, with 6% survival at 24 h (1/16 mice) and 0% survival at 48 h ($P = 0.005$ versus hA20(Δ-UP) group; Fig. 3c, d). Thus, absence of *TNFAIP3* downstream regulatory sequences has important pathophysiological consequences in vivo.

Autoantibodies to intracellular proteins and nucleic acids are a hallmark of autoimmune diseases. Several single-nucleotide polymorphisms near the hA20 (*TNFAIP3*) gene are independently associated with susceptibility to SLE[12–14]. In addition, mice deficient in A20 in B cells develop humoral autoimmunity in old age[31–33]. To determine whether decreased hA20 expression in hA20(Δ-DOWN) mice is associated with autoimmunity, we characterized the autoantibody profile of these mice. Aged hA20 (Δ-DOWN) mice (6–7 months old) exhibited spontaneous development of statistically significant elevations in anti-nuclear antibodies (ANAs) relative to controls (Supplementary Fig. 7b, c); ANAs were not significantly different from positive control SLE mice (Supplementary Fig. 7d). These results suggest that the decrease in hA20 expression observed in hA20(Δ-DOWN) mice is sufficient to predispose to development of autoimmunity over time.

As A20-deficient mice can get arthritis[34–36], we monitored the front and hind paws of CTR and Δ-DOWN mice weekly for paw swelling and deformity. Δ-DOWN mice showed a progressive development of paw swelling and redness beginning at 6 months of age with the incidence reaching ~84% as mice were monitored up to 12 months of age (Fig. 3e, upper panels, Fig. 3f). Histopathological analysis revealed inflammatory arthritis and dactylitis with synovitis involving the digits of all paws (Fig. 3e, lower panels). The bone, articular cartilage, synovium, and peri-articular soft tissues were infiltrated by mononuclear cells. Other tissues broadly evaluated at necropsy did not show any notable changes. To evaluate whether mRNA expression of inflammatory cytokines or chemokines was elevated in parallel with arthritic changes, RNA was extracted from paws of CTR and Δ-DOWN mice and Il1b, Cxcl9, and Cxcl10 transcripts were measured by quantitative PCR (qPCR). As shown in Fig. 3g, expression of these transcripts was upregulated in inflamed paws from Δ-DOWN mice. These findings are reminiscent of the arthritic

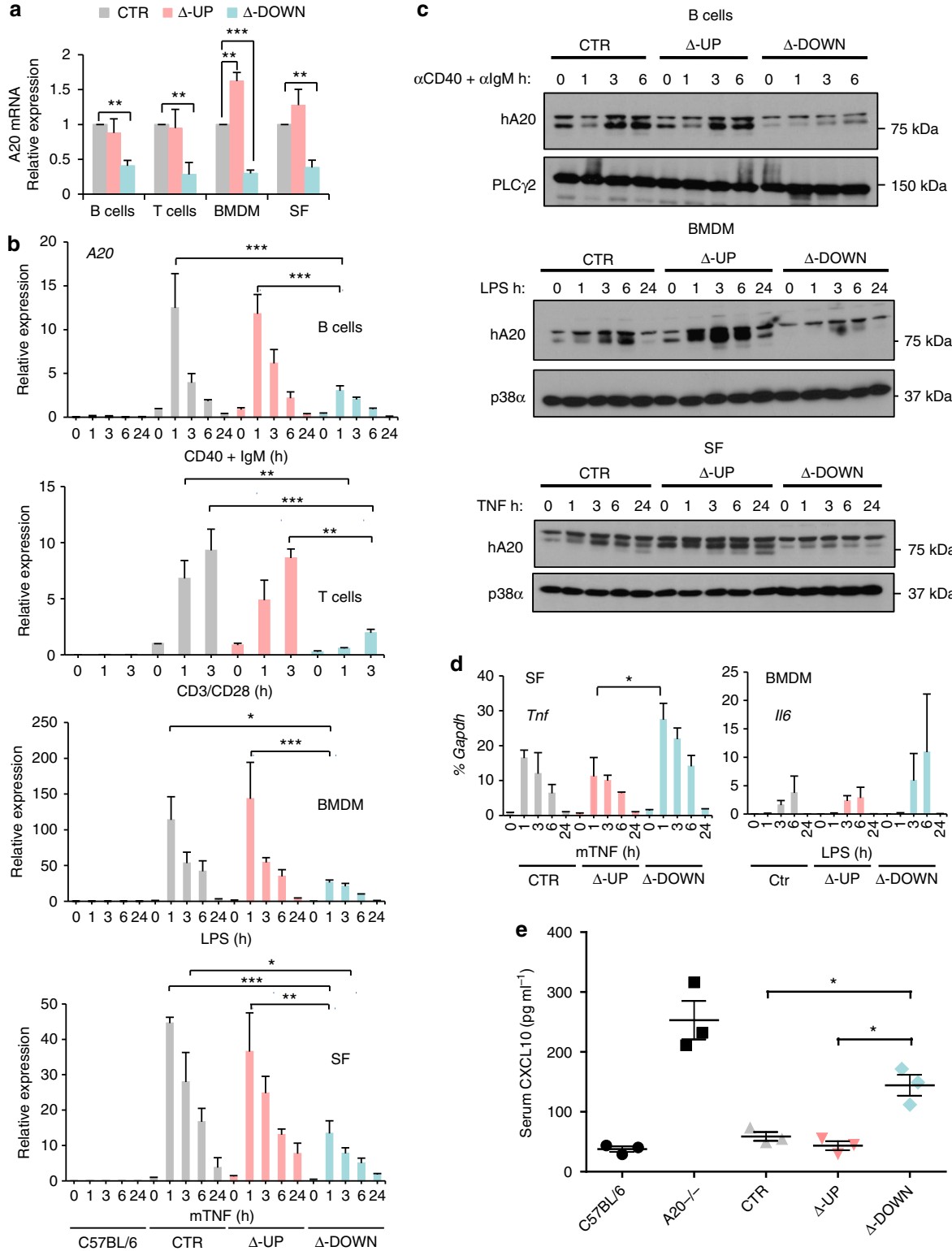

phenotype of mice with myeloid A20 deficiency[34] and suggest that the lower amounts of A20 expression observed in Δ-DOWN mice are sufficient for development of this inflammatory phenotype. Overall, the results indicate that the downstream sub-TAD containing four putative enhancers plays an important role in regulating hA20 expression in vivo and preventing development of inflammatory pathology and autoimmunity.

**Analysis of DE4 and TT>A enhancers using genome editing.** We wished to define the role of individual enhancers in the functionally important downstream sub-TAD. This region contains two putative monocyte-specific enhancers (DE2 and DE3), one enhancer (DE4) with highly open chromatin, positive histone marks, and transcription factor (TF) binding in monocytes, B cells, T cells, and fibroblasts, and a region of weakly open chromatin in B cells just upstream of DE3 that harbors a TT>A SNV previously linked with and suggested to be causally associated with SLE[14,37] (hereafter termed the "TT>A enhancer"; Figs. 1b and 4a; ENCODE data). The risk TT allele disrupts transcription factor binding to a region that functions as an enhancer in cell lines. We targeted DE4 based upon its chromatin state and presence in multiple relevant cell types, and the TT>A enhancer based upon pathophysiological importance. CRISPR-mediated deletion of both DE4 alleles, as described in Methods, did not affect A20 expression in 293T cells (Fig. 4b). To gain insight into DE4 function in primary immune cells, we generated transgenic mice in which DE4 was deleted in oocytes from hA20(CTR) mice using CRISPR technology (Fig. 4c and Supplementary Fig. 8); this results in mice identical to hA20(CTR) mice except for deletion of DE4. Surprisingly, deletion of DE4 in three independent CRISPR mouse lines minimally affected basal or LPS-induced hA20 mRNA or protein expression in B cells, T cells, BMDMs, and SFs, with only a slight and not significant trend toward lower mRNA at early time points after LPS stimulation of BMDMs (Fig. 4d). hA20 protein amounts also showed minimal decreases in DE4 mice (Fig. 4e, Supplementary Fig. 11b and 16). These results suggest redundant function of DE4 in the cells analyzed, and prompted analysis of an additional enhancer.

We chose to next target the TT>A enhancer based upon its linkage with autoimmune diseases[14,37]. We first deleted the TT>A enhancer (~653 bp, genomic location depicted in Fig. 4a) using recombineering technology (Supplementary Fig. 9 and 10a, b) and analyzed effects on *TNFAIP3* expression in various cell types. Interestingly, deletion of the TT>A enhancer resulted in a significant and strong reduction (by 60%) of basal and inducible hA20 mRNA in B and T lymphocytes (Fig. 5a, b). In contrast, in BMDMs and SFs there was a lesser reduction in hA20 mRNA amounts in hA20(ΔTT>A) relative to hA20(CTR) cells, and this reduction was only apparent at the 1 h time point (Fig. 5b). Accordingly, deletion of the TT>A enhancer strongly decreased hA20 protein expression in B cells, with a small effect in BMDM and SF (Fig. 5c, Supplementary Figs 11c, d, 12, 15c, and 17). To

address possible position effects, we also generated a mouse line in which the TT>A enhancer was deleted in oocytes from hA20 (CTR) mice using CRISPR technology as described above for the DE4 deletion. The CRISPR-mediated deletion of the TT>A enhancer region is 142 bp larger than the recombineering-mediated deletion because of constraints imposed by a requirement for a PAM sequence (Supplementary Figures 10c–e and Tables 5 and 6). CRISPR-mediated deletion of the TT>A enhancer in the context of, and at same genomic location as, the hA20(CTR) BAC significantly decreased hA20 mRNA expression in B cells (Fig. 5d), thus providing further support for the functional importance of this enhancer. Moreover, A20 expression in 293T cells with homozygous deletion of the TT>A enhancer was significantly decreased after TNF stimulation (Fig. 5e and Supplementary Fig. 10f). Collectively, the results demonstrate a function of the TT>A enhancer, which appears most prominent in B cells.

**Deletion of TT>A enhancer results in inflammatory arthritis.** To investigate the functional consequences of TT>A enhancer deletion, we analyzed cytokine expression after activation of lymphocytes. IL-6 and interferon (IFN)-γ mRNA amounts were significantly increased in activated B and T cells from Δ-TT>A mice relative to CTR mice (Fig. 5f). Analysis of in vivo phenotype revealed that both female and male Δ-TT>A mice had significantly lower weights than CTR mice at 12 months of age (Fig. 6a). Similar to Δ-DOWN mice, Δ-TT>A mice exhibited a progressive development of paw swelling and redness. However, in accord with a lesser effect of TT>A enhancer deletion on hA20 expression, arthritis onset was delayed and with lower incidence than in Δ-DOWN mice, with onset at 8 months of age and incidence reaching ~60% as the Δ-TT>A mice were followed to 12 months of age (Fig. 6b). Histological analysis of the paws of these mice showed arthritis and severe dactylitis with marked synovial and peri-articular inflammation, and with infiltration by mononuclear cells, osteoclasts, and bone erosion (Fig. 6c and Supplementary Fig. 13a). Immunohistochemical analysis of inflamed paws revealed infiltration with neutrophils, macrophages, B cells, and T cells, associated with deposition of immunoglobulin G (IgG; Fig. 6c and Supplementary Fig. 13a). Accordingly, *Cxcl9*, *Cxcl10*, *Il1b*, *Il6*, and *Tnf* mRNA amounts were significantly upregulated in inflamed paws from Δ-TTA mice, whereas *Il23a* transcripts were similar in both groups (Fig. 6d). In addition to immune cell infiltration locally at the clinically apparent site of paw inflammation, we noted expansion of CD11b+ and CD11b+Gr1+ myeloid cells in spleens of Δ-TT>A mice (Fig. 6e, f), which is in accord with expansion of the myeloid compartment observed in inflammatory arthritis models[34]. Finally, Δ-TT>A mice showed a trend toward higher CXCL10 and KC chemokines in serum, and ANA positivity (Supplementary Fig. 13b, c). Collectively, the results show that deletion of the TT>A enhancer at the *TNFAIP3* locus results in

**Fig. 2** Deletion of downstream sub-TAD compromises hA20 expression and results in elevated cytokine production. Primary cells were isolated from BAC transgenic mA20-deficient mice. **a**, **b** qPCR analysis of hA20 mRNA in transgenic primary cells. Primary B and T cells were isolated from spleens, macrophages cultured from bone marrow (BMDM), and synovial fibroblasts (SFs) were isolated from ankle joints. B cells were stimulated with antibodies against CD40 (2 μg ml⁻¹) and IgM (5 μg ml⁻¹), T cells with anti-CD3/anti-CD28-coated beads (1:1 cell to bead ratio), BMDM with LPS (25 ng ml⁻¹), and SF with TNF (20 ng ml⁻¹) for the indicated times. Data are combined from three independent experiments. Error bars indicate mean ± s.e.m., statistical significance was determined using one-way ANOVA followed by Tukey's multiple comparison test. *P < 0.05; **P < 0.01; ***P < 0.001. **c** Immunoblot analysis of hA20 protein in primary cells from transgenic mice. Primary cells were obtained and stimulated as in **b**. Data are representative of two to three experiments. **d** qPCR analysis of cytokine mRNA from ex vivo-stimulated cells. Data are combined from two independent experiments. Error bars indicate mean ± s.e.m., statistical significance was determined using one-way ANOVA followed by Tukey's multiple comparison test, *P < 0.05. **e** Luminex analysis of CXCL10 protein in serum of transgenic and control mice. Data from three individual mice are shown. Error bars indicate mean ± s.e.m., statistical significance was determined using one-way ANOVA followed by Tukey's multiple comparison test, *P < 0.05

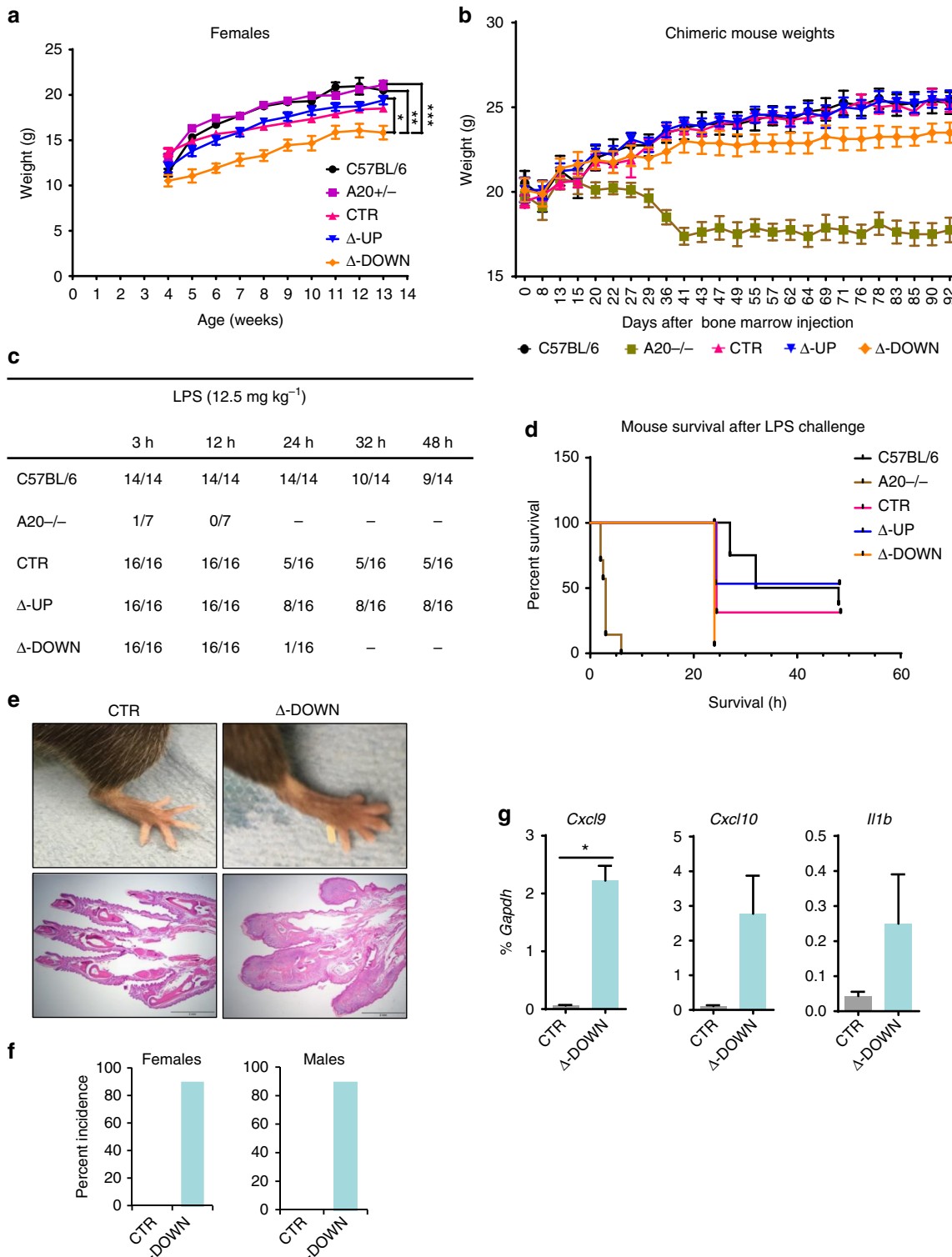

**Fig. 3** Downstream enhancers prevent development of inflammatory pathology and autoimmunity. **a** Growth curves of BAC transgenic and control mice. $n \geq 10$ per Tg group. **b** Growth curves of C57BL/6 mice chimeric for bone marrow from BAC transgenic or *Tnfaip3*−/− mice. $n = 9$ per group. Error bars indicate mean ± s.e.m., statistical significance was determined using one-way ANOVA followed by Tukey's multiple comparison test, *$P < 0.05$; **$P < 0.01$; ***$P < 0.001$. **c**, **d** Survival data for LPS-challenged mice shown in tabulated form (**c**) or as Kaplan–Meier plot (**d**). Data pooled from two experiments, $n = 16$ for transgenic mice. **e** Photographs of hind paws of CTR and Δ-DOWN mice at the age ≥ 6 months (top). Photomicrographs of H&E-stained sections of paws and digits; scale bars, 2 mm (bottom). **f** The cumulative incidence of arthritis in CTR ($n = 5$ for females and 7 for males) and Δ-DOWN ($n = 8$ for females and 5 for males) mice starting at 6 months and monitored up to 12 months of age. **g** mRNA levels of *Cxcl9*, *Cxcl10*, and *Il1b* in cells isolated from hind paws of CTR ($n = 3$) and Δ-DOWN ($n = 3$) mice at the age ≥ 6 months. Results are shown as mean ± s.e.m. *$P < 0.05$, paired *t*-test

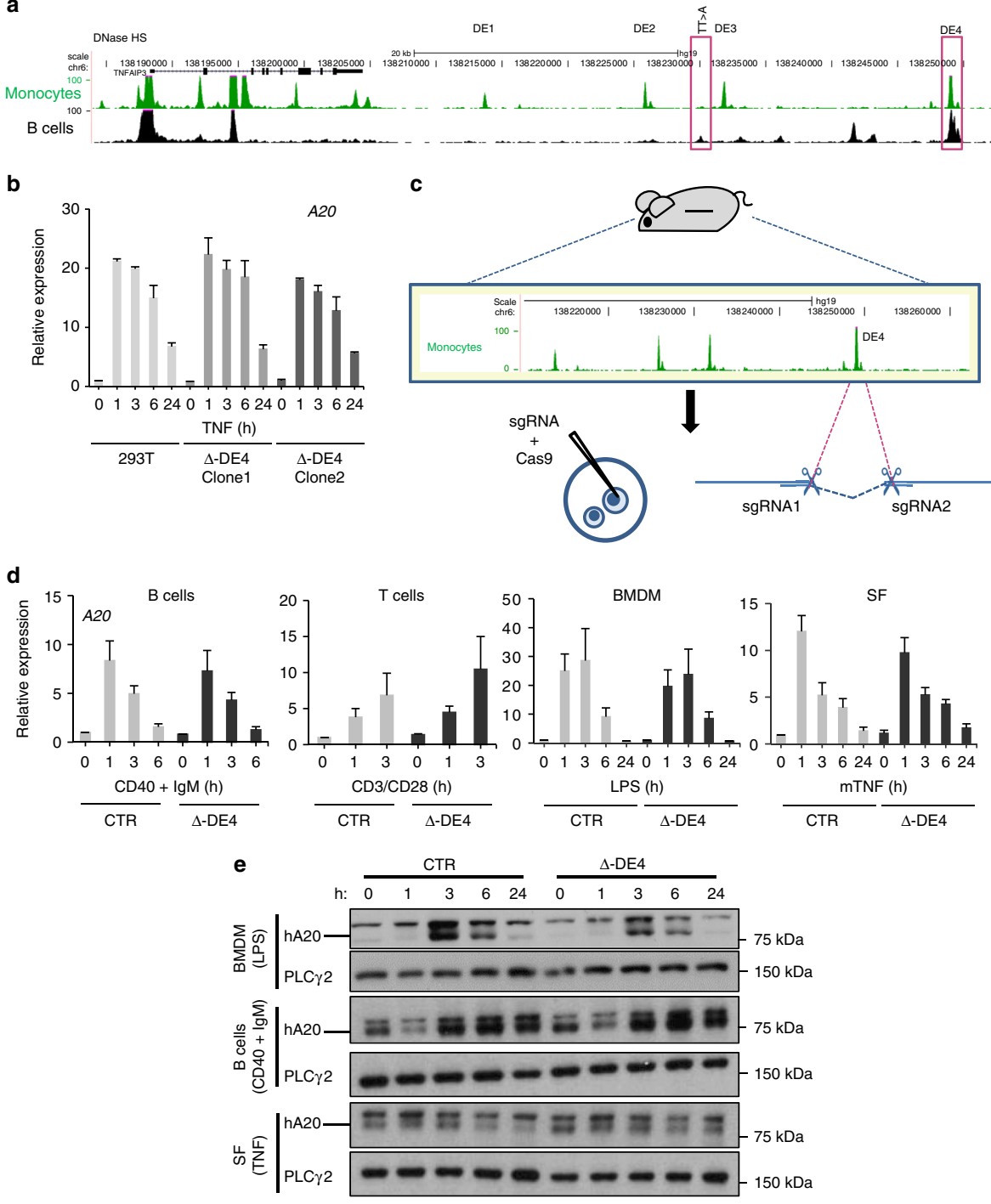

**Fig. 4** Genome editing of DE4 enhancer and functional analysis in BAC transgenic mice. **a** Gene tracks showing genomic locations of downstream enhancers. **b** CRISPR-mediated deletion of downstream enhancer 4 (DE4) does not affect hA20 expression in 293T cells. qPCR analysis of hA20 mRNA in control and DE4-deleted 293T cells, error bars represent mean ± s.e.m. **c** Strategy of CRISPR-mediated deletion of DE4 in oocytes from hA20(CTR) mice. **d** qPCR analysis of hA20 mRNA in B cells, T cells, BMDM, and SF. Primary B cells were isolated from spleens, macrophages cultured from bone marrow (BMDM), and synovial fibroblasts (SFs) were isolated from ankle joints. B cells were stimulated with antibodies against CD40 (2 µg ml$^{-1}$) and IgM (5 µg ml$^{-1}$), BMDM with LPS (25 ng ml$^{-1}$), and SF with TNF (20 ng ml$^{-1}$) for the indicated times. Pooled data from three independent experiments with three different Δ-DE4 mouse lines is shown. Error bars indicate mean ± s.e.m., statistical significance was determined using one-way ANOVA followed by Tukey's multiple comparison test. **e** Immunoblot analysis of hA20 protein expression in primary cells obtained and stimulated as in **d**. Representative of two independent experiments

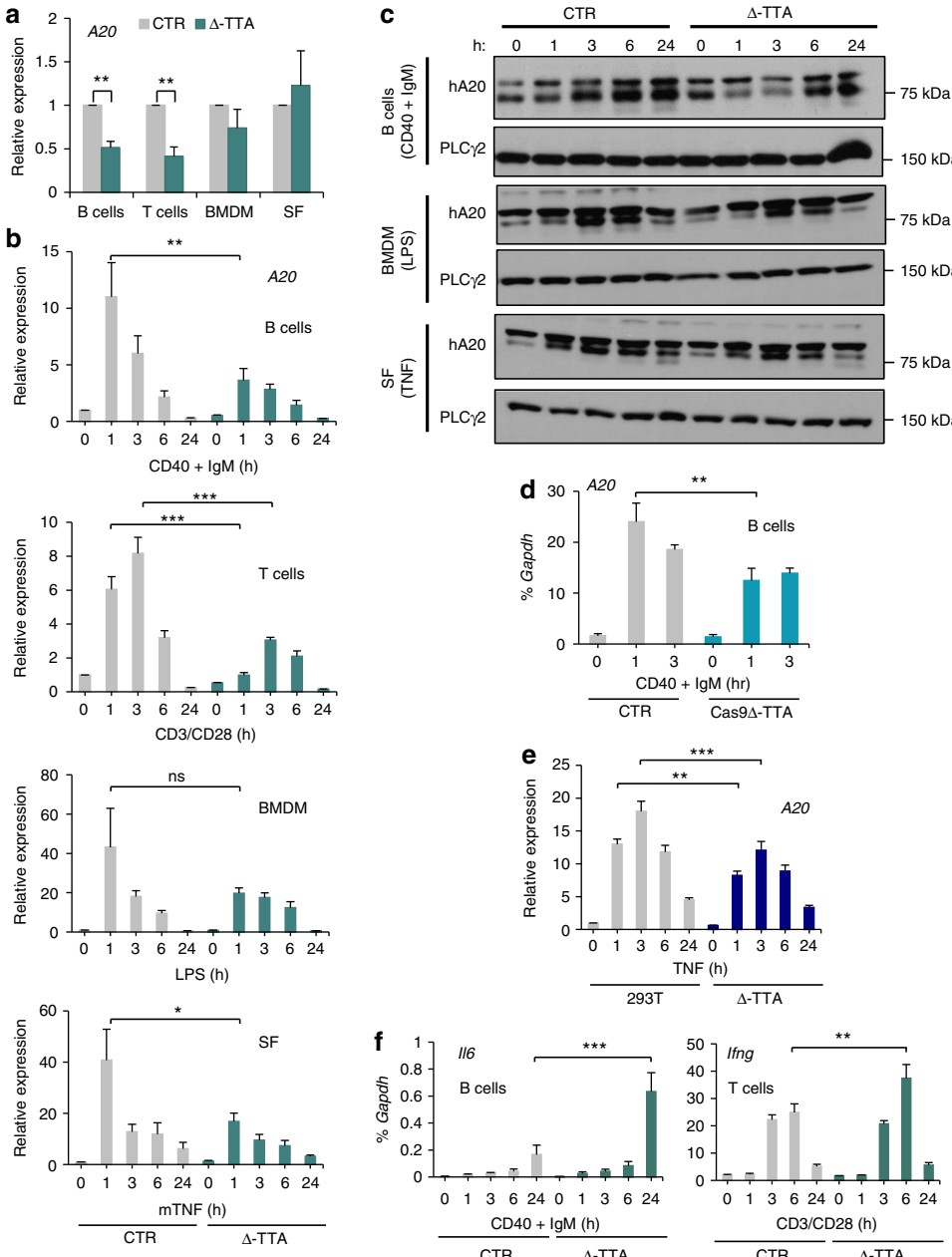

**Fig. 5** Genome editing in vivo reveals cell-specific and signal-responsive functions of the TT>A enhancer. **a**, **b** qPCR analysis of basal (**a**) or inducible (**b**) hA20 expression in cells from BAC transgenic mice in which TT>A enhancer was deleted using recombineering. Primary B and T cells were isolated from spleens, macrophages cultured from bone marrow (BMDM), and synovial fibroblasts (SFs) were isolated from ankle joints. B cells were stimulated with antibodies against CD40 (2 μg ml$^{-1}$) and IgM (5 μg ml$^{-1}$), T cells with anti-CD3/anti-CD28-coated beads (1:1 cell to bead ratio), BMDM with LPS (25 ng ml$^{-1}$), and SF with TNF (20 ng ml$^{-1}$) for the indicated times. Data are combined from three independent experiments. Error bars indicate mean ± s.e.m., statistical significance was determined using one-way ANOVA followed by Tukey's multiple comparison test. *$P < 0.05$; **$P < 0.01$; ***$P < 0.001$; ns, not significant. **c** Immunoblot analysis of hA20 protein expression. Representative of two to four experiments. Primary cells were obtained and stimulated as in **b**. **d** qPCR analysis of hA20 mRNA in B cells isolated from mice bearing a CRISPR-mediated deletion of TT>A enhancer. Error bars indicate mean ± s.e.m. of four independent experiments, statistical significance was determined using one-way ANOVA followed by Tukey's multiple comparison test. **$P < 0.01$. **e** CRISPR-mediated deletion of TT>A enhancer affects hA20 expression in 293T cells. Error bars indicate mean ± s.e.m. of five independent experiments, statistical significance was determined using one-way ANOVA followed by Tukey's multiple comparison test. **$P < 0.01$; ***$P < 0.001$. **f** qPCR analysis of cytokine mRNA from ex vivo-stimulated B and T cells. Error bars indicate mean ± s.e.m. of three independent experiments, statistical significance was determined using one-way ANOVA followed by Tukey's multiple comparison test. **$P < 0.01$; ***$P < 0.001$

an autoimmune/inflammatory phenotype most clearly evident as arthritis.

## Discussion

Enhancers are key regulators of gene expression. Allelic variants linked with complex diseases often lie within enhancers and have

been proposed to contribute to pathophysiology by altering gene expression[1]. Little is known about enhancers that regulate human disease-associated genes, especially about the importance of individual enhancers for in vivo and disease phenotypes. In this study we used a combination of BAC transgenesis and genome editing to identify enhancers important for expression of

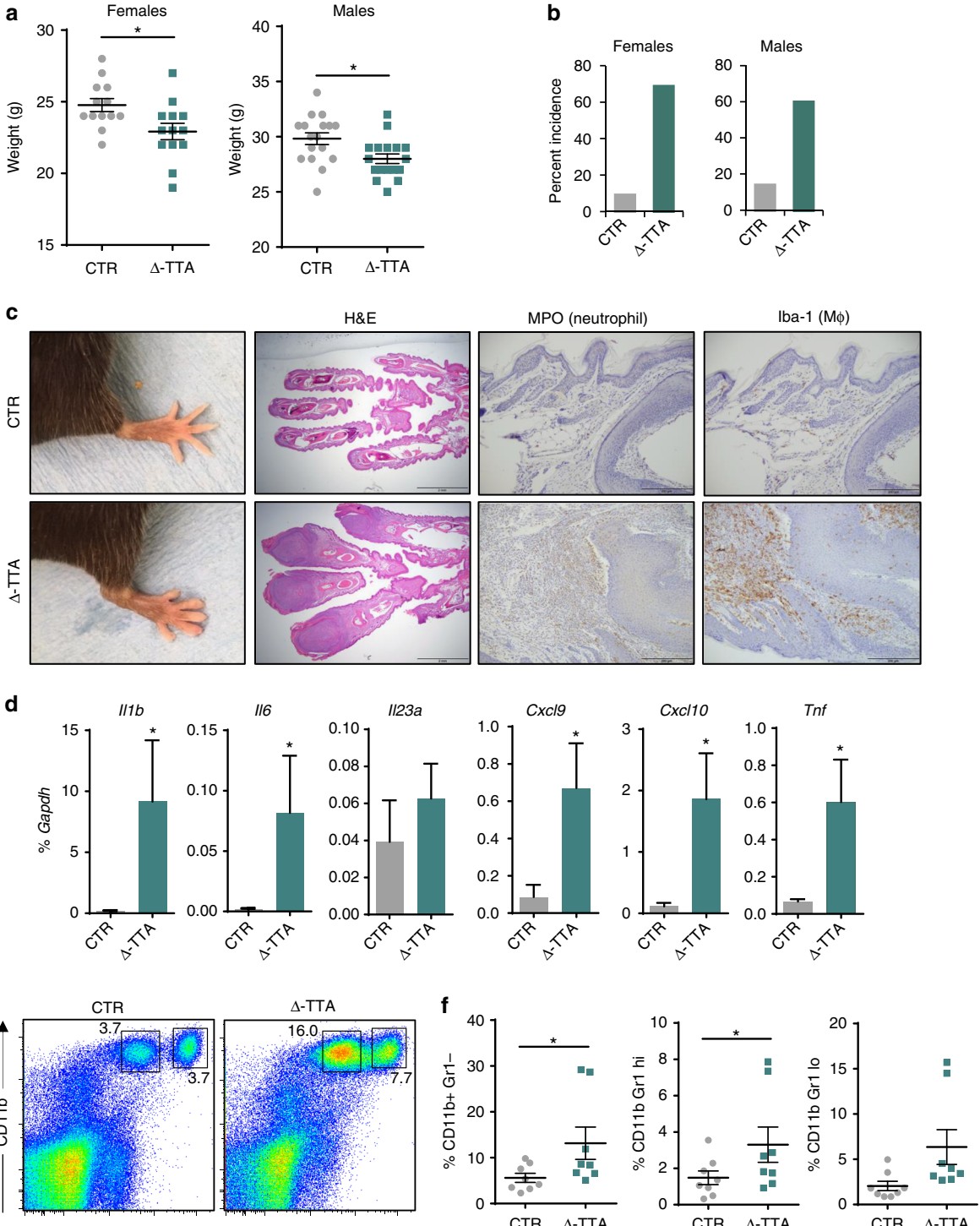

**Fig. 6** Δ-TT>A mice develop spontaneous arthritic phenotype with age. **a** Whole-body weights of CTR and Δ-TT>A mice at 12 months of age (*n* = 13 for females and 17 for males). Results are shown as mean ± s.e.m., statistical significance was determined using unpaired two-tailed *t*-test, *P < 0.05. **b** The cumulative incidence of arthritis in CTR (*n* = 14 for females and 25 for males) and Δ-TTA (*n* = 18 for females and 19 for males) mice ≥11 months of age. **c** Photographs of hind paws of CTR and Δ-TTA mice at ≥11 months of age; photomicrographs of sections of paws and digits stained with H&E (scale bars, 2 mm), myeloperoxidase (MPO), and Iba-1 (anti-Iba-1) showing increased infiltration of neutrophils and macrophages; scale bars, 200 μm. Photomicrographs are representative of 5 (CTR) and 6 (Δ-TT>A) mice. **d** mRNA levels of *Il1b*, *Il6*, *Il23a*, *Cxcl9*, *Cxcl10*, and *Tnf* in cells isolated from hind paws of CTR (*n* = 7) and Δ-TT>A (*n* = 7) mice at the age of ≥11 months. Results are shown as mean ± s.e.m., Wilcoxon matched-pairs signed rank test, *P < 0.05. **e** Flow cytometric analysis of CD11b+ and CD11b+Gr1+ cells from spleens of CTR and Δ-TT>A mice presented as **e** representative dot plots and **f** percentage of cells. Results are shown as mean ± s.e.m. Statistical significance was determined using Wilcoxon matched-pairs signed rank test, *n* = 8, *P < 0.05

autoimmunity-associated *TNFAIP3*, which encodes the key negative regulator of immune cell activation A20, in primary immune cells and in vivo. Our results identified a genomic region spanning four regulatory elements that is important for *TNFAIP3* expression in immune cells and is located within a *TNFAIP3*-associated sub-TAD. Importantly, deletion of this sub-TAD results in an in vivo hyper-inflammatory and autoimmune phenotype. Genome editing of individual enhancers revealed cell type-specific and signal-responsive enhancer function, and provided strong evidence for the functional importance of an enhancer harboring a SLE-associated TT>A allelic variant in immune cells, including suppression of autoimmunity and arthritis in vivo. These findings reveal insights into enhancer-mediated regulation of the key autoimmunity-associated human *TNFAIP3* gene, and to our knowledge represents the first demonstration of a direct and causal link between human enhancers and in vivo disease phenotypes.

Regulatory sequences and enhancers are less conserved than coding sequences, and thus direct analysis of human genes is required to fully understand regulation of gene expression and the functions and mechanisms of action of disease-associated variants that lie within enhancers[18]. However, investigation of human gene-enhancer function has been limited by the need to use cell line models that often do not faithfully recapitulate gene regulation and enhancer function, most likely because cell transformation-associated pathways alter the epigenome and chromosomal context, and dysregulate signaling pathways. For example, we have observed dramatic differences in the magnitude and kinetics of gene expression in monocytic cell lines such as THP-1 cells relative to primary monocytes, which appears related at least in part to epigenomic differences. In addition, ethical and technical considerations clearly preclude the ability to assess function by altering human enhancers in vivo.

We have addressed these limitations by using a BAC transgenic approach that generally faithfully recapitulates human gene expression in transgenic mice by introducing extensive regulatory information that confers position-independent gene expression, where genes are regulated by their native promoters and enhancers in a chromatinized and insulated context[18]. Strikingly, even though regulatory elements may be less conserved than coding sequences, human BACs are typically faithfully expressed in BAC transgenic mice, suggesting conservation of regulatory logic[18,19], even at the whole chromosome level[16]. We have taken advantage of this BAC transgenic system to identify function of context-dependent enhancers in primary immune cells. Equally importantly, we were able to show the in vivo role of these enhancers in suppressing inflammation and autoimmunity. However, our work remains subject to caveats related to possible position effects secondary to BAC integration sites, and to faithful expression of human genes in mouse cells. We have experimentally addressed the position effect issue by analyzing the effects of deleting the sub-TAD and the TT>A enhancer in independent mouse lines; in the case of the TT>A enhancer this included deletion in the context of the control CTR-BAC, thus at the same genomic location. Concerning expression of human genes in mouse cells, our analysis of hA20 gene expression patterns and chromatin accessibility of a limited number of enhancers supported faithful regulation of the hA20 gene in mouse immune cells. This is in accord with a substantial body of literature showing that functional conservation of transcriptional regulation between human and mouse is considerably greater than sequence conservation amongst regulatory elements[18,38]. Nonetheless, it will be important in future work to more fully analyze the numerous hA20 gene enhancers and perform detailed epigenomic analysis of BAC transgenes to fully understand regulatory mechanisms.

The "TT>A" allelic variant (tandem polymorphic dinucleotides rs148314165 and rs200820567; the risk allele is termed TT>A) located 41.6 kb downstream of the *TNFAIP3* TSS is located within the most highly replicated ≈100 kb risk haplotype for autoimmune diseases. The TT>A risk allele was proposed to be causally associated with SLE based on decreased NF-κB binding and associated lower A20 expression[14]. This SLE-associated variant TT>A was shown to be located in an enhancer in cell lines[37]. We have now provided the most compelling evidence for the functional importance of this enhancer by showing its importance in primary immune cells and that deletion of this enhancer results in decreased hA20 expression in vivo. Our work sets the stage for future dissection of this enhancer, including using genome editing to introduce the TT>A risk allele and determine functional consequences in vivo. This approach can be extended to analysis of additional disease-associated SNVs and could lead to the development of more relevant preclinical models based upon incorporation of human genetic variants that actually cause disease. In addition, identifying the most functionally important enhancers of human disease-associated genes, and the signaling pathways that regulate them, opens new approaches to therapeutics by targeting these enhancers with pharmaceutical or genome editing approaches to modulate pathogenic gene expression.

A20 loss-of-function mutations result in a Behcet's syndrome-like phenotype[11], whereas genome-wide association study-identified SNVs in noncoding regions of the *TNFAIP3* locus are associated with a variety of distinct autoimmune diseases, including SLE, RA, inflammatory bowel disease (IBD), type I diabetes mellitus, and psoriasis. The reasons why these SNVs are linked with various distinct disease phenotypes are not well understood, but possible explanations include differences in genetic background (the sum total of a patient's disease-predisposing SNVs) and environmental exposures, including effects of microbiota. Initial insights into distinct disease phenotypes have been provided by conditional knockout studies in mice, which have demonstrated that cell type-specific deletion of A20 results in distinct pathologies. Most relevant for this study, deletion of A20 in B cells results in a mild autoimmune syndrome with increased germinal center formation and autoantibody production; in dendritic cells results in IBD, spondyloarthritis and a protean autoimmune syndrome that resembles SLE; and in myeloid cells results in erosive inflammatory arthritis involving the paws[33–36,39]. Although these studies provide important insights into the relationship between A20 function in specific cell types and disease phenotypes, complete A20 deficiency in gene knockout models does not model well the in vivo effects of subtle modulation of A20 expression by disease-associated noncoding SNVs, which also can have cell- and stimulus-specific effects on gene expression. In this regard, our work provides among the first insights into disease phenotypes associated with genetic changes in regulatory elements (enhancers) that relatively modestly affect A20 expression in several cell types, which mirrors what occurs in human diseases.

Although both Δ-DOWN and Δ-TT>A mice exhibit autoimmune phenomena, perhaps surprisingly the predominant disease manifestation in both mouse lines was inflammatory arthritis and dactylitis (inflammation of digits) that mimics the arthritic phenotype of mice with myeloid-specific A20 deletion[34,35], except it is more anatomically restricted (myeloid A20-deleted mice also have involvement of ankle joints). Dactylitis is most commonly observed in autoimmune psoriatic arthritis, although it has been observed in RA. In the context of the Δ-DOWN mice, these findings indicate that small decreases in A20 expression in several cell types (lymphocytes, myeloid cells, and SFs) can phenocopy complete deletion in one cell type, namely myeloid cells. Most

likely functional cooperation amongst several cell types with modest decreases in hA20 expression and thus increased cell activation potential results in a significant clinical phenotype in Δ-DOWN mice. Thus, broad defects in A20 expression in multiple cell types can result in a specific and focused clinical phenotype under specific experimental conditions. Interestingly, the more modest effects of TT>A enhancer deletion on A20 expression, which were mostly restricted to lymphocytes, resulted in a similar disease phenotype, although with later onset and decreased incidence. This, together with T-cell infiltration of synovium, suggests a key role for T cells in pathogenesis, which will need to be tested in future work. The selective development of spontaneous arthritis in the paws and digits in our study may be related to long-term mechanical stress related to weight bearing, and to low A20 expression in the relevant tissue (stromal) cells, in this case SFs. It will be interesting to test in future work how differences in mouse strain, genetic background, and environmental stressors and microbiota, influence disease phenotypes, which could yield insights into variable disease phenotypes associated with human SNVs and mutations.

Our results identify signal-responsive enhancers and are in accord with emerging models of hierarchical enhancer function[2,3], but also show massive redundancy amongst enhancers, particularly the large number of enhancers upstream of the *TNFAIP3* gene body. In addition, in contrast to previous studies, enhancer strength did not correlate with chromatin state, as the functionally potent TT>A enhancer linked with autoimmune diseases exhibited relatively low chromatin accessibility (Figs. 1b and 5a), although this enhancer does show substantial TF binding in various cell types (Supplementary Fig. 14). In contrast, the DE4 enhancer that clearly satisfied enhancer criteria based on chromatin accessibility and inducible histone acetylation exhibited minimal function, possibly secondary to redundancy at this complex multi-enhancer locus. These findings highlight the importance of functional analysis in addition to epigenomic analysis, analysis of TF binding, and of broad systematic enhancer screens at disease-associated gene loci[40,41], ideally performed in primary cells relevant for disease pathogenesis. Our humanized transgenic model provides a platform for dissecting tissue-specific enhancers and understanding regulation of *TNFAIP3*, and for identifying functionally important SNVs that are causally related with autoimmune diseases. The discovery of a functionally important sub-TAD and enhancer at the *TNFAIP3* locus will guide and focus future work to identify additional functional enhancers and SNVs. These results will be important in determining how to therapeutically manipulate A20 expression to suppress autoimmunity.

## Methods

**BAC clones and generation of transgenic mice.** Three BAC clones RP11-10J5 (CTR-BAC), RP11-953L22 (Δ-UP-BAC), and CTD-2657E11 (Δ-DOWN-BAC) were obtained from Children's Hospital Oakland Research Institute. Experiments were approved by the Institutional Animal Care and Use Committee of Hospital for Special Surgery and Weill Cornell Medical College. BAC DNA was prepared using cesium chloride gradient centrifugation, linearized with PI-SceI, purified through CL4B columns, 18 fractions from the flow through were collected and detected on Pulsed Field Gel and stored at 4 °C. The purified BAC DNA was then microinjected into C57BL/6 fertilized embryos, and implanted in pseudopregnant female mice. Out of >135 mice screened we obtained 12 lines with germline transmission of structurally intact *TNFAIP3* as assessed by Southern blotting. C57BL/6J mice were purchased from Jackson Laboratory. These founder mice were later genotyped by PCR. Six human-specific primers spanning the *TNFAIP3* gene and surrounding sequences were designed in order to characterize founders carrying the entire construct and for PCR genotyping (Supplementary Table 3) and the transgenic BACs were confirmed by Southern blot analysis (Supplementary Fig. 2b).

**Determination of transgene copy number.** Expression of BAC transgenes typically correlates with transgene copy number[42–44]. Transgene copy number was

determined by Taqman digital PCR using a human *TNFAIP3*-specific primer obtained from Thermo Fisher Scientific/Life Technologies. Genomic DNA from transgenic lines was digested with *Eco*RI (New England Biolabs) and 20 ng of this digested DNA was analyzed by QuantStudio 3D Digital PCR System (Life Technologies, Carlsbad, CA, USA) to determine the transgene copy number. Analysis was done using the QuantStudio AnalysisSuite cloud software. Transferrin receptor gene is used as an internal control, which has two copies in all mouse lines (Supplementary Fig. 2c).

**Mouse-breeding strategy.** Founders with one copy of human BAC transgenes (termed hA20(CTR), hA20(Δ-UP), and hA20(Δ-DOWN) were chosen for further breeding for most experiments. These mice were crossed with heterozygous *Tnfaip3*+/- mice that had been previously obtained from Dr. Averil Ma (University of California, San Francisco) to yield hemizygous transgenic mice on heterozygous *Tnfaip3* KO background. These mice were backcrossed with heterozygous *Tnfaip3* KO mice to obtain hemizygous transgenic mice on homozygous *Tnfaip3* KO background (Tg+ *Tnfaip3*–/–). Tg+ *Tnfaip3*–/– mice were crossed to obtain progeny that, after genotyping, were used for experiments. Age- and gender-matched controls were used, except for *Tnfaip3*+/− mice, which die postnatally (approximately at 4 weeks in our colony) and were use at 3–4 weeks of age. Genotyping was conducted to analyze the transgenes and *Tnfaip3* (KO) alleles. To genotype mice, tails were clipped and tissue was digested overnight in a tail digestion solution (50 mM Tris, pH 8.0; 100 mM NaCl; 20 mM EDTA; 5% SDS; and 100 µg ml⁻¹ Proteinase K (Roche Diagnostics, Cat. # 3115828001)). Genotyping of founder mice was performed as described above. First litters of offspring mice were genotyped using the same set of primers and Southern blot analysis, but once BAC integrity in F1 mice was verified, three primer sets were used for genotyping to analyze the transgene and KO alleles (Supplementary Table 4). The first set measures the human *TNFAIP3* gene. The second set is for neomycin cassette sequence, which integrates into the mA20 locus. The third primer set measures the status of mouse *Tnfaip3* gene.

Mice were originally housed under standard SPF conditions in the HSS animal facility; the entire HSS mouse colony was moved to the Weill Cornell Medicine animal facility in 2016. The in vivo arthritis phenotype was observed in both animal facilities. The animal protocols were approved by Institutional Animal Care and Use Committee at both institutions, HSS and WCM. In initial experiments at HSS, the hA20(Δ-DOWN) line did not breed well, which appeared related to inflammation in the uterine lining possibly related to commensal flora, consistent with previous literature. Issues with breeding resolved when mice were treated with doxycycline. To control the comparison of the different mouse lines, breeding cages for all lines were maintained on doxycycline. For the experimental groups, pups were weaned at 3 weeks of age, and switched to a regular rodent chow (except for the mice used in experiments in Fig. 5a–c, f, which were maintained on antibiotics post weaning). For the growth curve experiments, mice that were tested to be hemizygous for the hA20 transgene and mouse A20 null (Tg+/− *Tnfaip3* –/−) were weighed weekly at the same hour on the same day of the week. C57BL/6 mice and A20+/− were also weighed as controls. In Fig. 3a, mice were randomly selected from our mouse colony; because the number of mice per line is subject to litter quantity and size, groups were of unequal size. Animals that died were not excluded from the experiment. In the growth curve for bone marrow chimeric mice (Fig. 3b) a cohort of mice was generated and followed.

**Preparation and culture of murine T and B lymphocytes.** The spleen was removed and mechanically disrupted using wire mesh and a syringe. Cells were spun down in a 50-ml Falcon tube (5 min, 300 × g), pelleted cells were resuspended in MACS buffer (30 ml per spleen, Miltenyi Biotec), spun down (5 min, 300 × g), and cells were filtered through a fresh cell strainer. Cells were counted and murine B lymphocytes (B cells) were isolated from spleen using positive selection with anti-CD45R (B220) magnetic beads (Miltenyi Biotec) as directed in the manufacturer's protocol. B cells were cultured at 37 °C and 5% CO₂ in Roswell Park Memorial Institute (RPMI) 1640 medium supplemented with 10% fetal bovine serum (FBS), 5% penicillin-streptomycin (pen-strep), and 5% L-glutamine. The B cells were then stimulated with purified NA/LE tat anti-mouse CD40 antibody (2 µg ml⁻¹, BD Pharmingen, Cat. # 553787) and donkey anti-mouse IgM antibody (5 µg ml⁻¹, Jackson ImmunoResearch, Cat. # 715-006-020) and hA20 expression was analyzed at both the mRNA and protein levels. T lymphocytes (T cells) were isolated from the flow through of B-cell-positive selection. These cells were positively selected using anti-CD90.2 (Thy1.2) Microbeads (Miltenyi Biotec) and cultured in the same conditions as the B cells. Cells were stimulated with anti-CD3 and anti-CD28 antibodies (concentration based on vendor instructions, Life Technologies, Cat. # 11452D) and collected at various time points indicated in the legends. The purity of cell populations was assessed by fluorescence-activated cell sorting. Pooled splenocytes from two mice were used for each group in each experiment.

**Isolation and culture of mouse BMDM.** Mice 6–10 weeks of age were euthanized with carbon dioxide (CO₂) followed by cervical dislocation and sterilization of femurs and tibias in 70% ethanol. To obtain the mouse BMDM, proximal and distal ends of the femurs and tibias were transversally cut and the bone marrow was flushed out by injecting 5 ml of Corning Dulbecco's Modified Eagle Medium

(DMEM) supplemented with 10% heat-inactivated FBS (Fisher Scientific), 5% pen-strep (Life Technologies), and 5% L-glutamine (Life Technologies) into the medullary canal with a syringe and 25-gauge needle. Cells were centrifuged and resuspended in DMEM and further supplemented with 10% L929 cell-conditioned medium. Cells were plated into four 10 cm diameter cell culture dishes (BD, Franklin Lakes, NJ) and cultured for 5 days at 37 °C and 5% $CO_2$. After reaching confluence, cells were lifted with gentle scraping, centrifuged, and split into six-well culture dishes and stimulated with the TLR4 ligand LPS (Invivogen, Cat. # TLRL-3pelps).

**Preparation of SFs.** After euthanization, mice were sterilized in 70% ethanol. Preparation of SFs was done as previously described[45]. The front- and hindlimbs were collected and washed in DMEM-F12 (Corning, Corning, NY, USA) supplemented with 10% heat-inactivated FBS, 100 U ml$^{-1}$ penicillin, and 100 mg ml$^{-1}$ streptomycin. All further dissection was performed with tissues immersed in culture media. Attached skin, nail, muscle, and tendon were removed. Synovial tissue were incubated in 20 ml of culture media containing 1 mg ml$^{-1}$ of collagenase type 4 (Worthington Biochemical Corp., Freehold, NJ, USA; 220 U mg$^{-1}$) with gentle rocking for 1–2 h at 37 °C and filtered through a 100 micron cell strainer (Falcon). Cells were spun down at $1100 \times g$ for 5 min and the cell pellet was resuspended in 10 ml of fresh media and cultured at 37 °C and 5% $CO_2$. Culture media was changed every 3 days and cells subcultured at 80–90% confluence prior to characterization at passage four. Upon characterization, FLS were simulated with recombinant murine TNFα (20 ng ml$^{-1}$, Peprotech, Cat. # 315-01A) and collected for mRNA and protein analysis.

**Real-time qRT-PCR.** RNA was extracted using RLT buffer (Qiagen) and treated with DNase I (Qiagen). cDNA was synthesized using the RevertAid First Strand cDNA Synthesis Kit (Thermo Fisher Scientific) and Real-time qRT-PCR was performed using Fast SYBRGreen Master Mix (Applied Biosystems) and a ABI7500 Fast system (ABI). Transcript levels were calculated relative to corresponding GAPDH (mouse or human) levels in each sample.

Primer sequences for quantitative reverse transcription-PCR were as follows:
mouse GAPDH F: 5′-ATCAAGAAGGTGGTGAAGCA-3′
mouse GAPDH R: 5′-AGACAACCTGGTCCTCAGTGT-3′
mouse Cxcl9 F: 5′-GAACTCAGCTCTGCCATGAAG-3′
mouse Cxcl9 R: 5′-GGCTTGGGGCAAACTGTTTGAGG-3′
mouse Cxcl10 F: 5′-CATGAACCCAAGTGCTGCCGTC-3′
mouse Cxcl10 R: 5′-CACGTGGGCAGGATAGGCTCGC-3′
mouse IL-1β F: 5′-CAGGCAGGCAGTATCACTCA-3′
mouse IL-1β R: 5′-CAGCTCATATGGGTCCGACAGC-3′
mouse IL-2 F: 5′-CCCACTTCAAGCTCCACTTCAAGC-3′
mouse IL-2 R: 5′-CATCCTGGGGAGTTTCAGGTTCC-3′
mouse IL6 F: 5′-GAGGATACCACTCCCAACAGACC-3′
mouse IL6 R: 5′-AAGTGCATCATCGTTGTTCATACA-3′
mouse IFN-γ F: 5′-GGATATCTGGAGGAACTGGC-3′
mouse IFN-γ R: 5′-GCGCCAAGCATTCAATGAGCTC-3′
mouse TNF F: 5′-CCACCACGCTCTTCTGTCTAC-3′
mouse TNF R: 5′-CCACTTGGTGGTTTGCTACGAC-3′
hA20 F: 5′-AGCATAGGGAGGGAGTGATAACTC-3′
hA20 R: 5′-GTGCTCTCCAACACCTGAAAAGG-3′

**Western blot analysis.** Western blotting was performed using standard methodology with the following antibodies: A20/TNFAIP3 (5630, Cell Signaling Technology) with 1:1000 dilution; PLC γ2 (sc-407, Santa Cruz Biotechnology) with 1:1000 dilution; and p38α (SC-535, Santa Cruz Biotechnology) with 1:1000 dilution.

**Cytokine detection.** Protein levels of CXCL10 and KC were measured (pg ml$^{-1}$) in duplicate using the multiplex mouse Cytokine/Chemokine Kit (MYCTOMAG-70K-04 Mouse cytokine Magnetic kit, Millipore Corp., Billerica, MA, USA) per the manufacturer's instructions. Briefly, the assay plate was washed with wash buffer for 10 min at room temperature. After decanting the wash buffer, standards, assay buffer (blank), or serum samples diluted in serum matrix were added to each well and incubated overnight at 4 °C on an orbital shaker with fluorescently labeled capture antibody-coated beads. After overnight incubation with capture antibodies to detect CXCL10 and KC (both antibodies were used at 1:50 dilution), well contents were removed via the washing instructions provided in the protocol. Biotinylated detection antibodies were then added to the wells and incubated with samples for 1 h at room temperature while shaking. After incubation, well contents were removed as previously described and streptavidin-phycoerythrin was added to each well for 30 min at room temperature. After the incubation period, samples were washed and resuspended in Sheath Fluid. Plates were run on the Luminex MagPix® machine and data were collected using the Luminex xPONENT® software (v. 4.2). Analysis of the cytokine/chemokine median fluorescent intensity was performed using the Milliplex® Analyst software (v. 5.1).

**Generation of bone marrow chimeras.** Bone marrow cells were isolated from 8- to 12-week-old male donor C57BL/6, mA20-deficient, hA20(CTR), hA20(Δ-UP),

and hA20(Δ-DOWN) mice by flushing the femurs and tibias with 5 ml ice-cold media (RPMI, 10 mM HEPES, 25 U ml$^{-1}$ heparin, and 5% FBS) using a 27.5-gauge needle. The cells were washed by centrifugation ($453 \times g$ at 4 °C for 10 min) and the cell pellet was resuspended in ice-cold media at $2.5 \times 10^7$ cells per ml. Recipient C57BL/6 mice (male, 6 weeks old, 9 per group) were sublethally irradiated with 875 RAD. Twenty-four hours after irradiation, they were transplanted intraorbitally with $5 \times 10^6$ cells. Mouse weights were recorded throughout and after the reconstitution process ($n = 9$), and mice were fed autoclaved food in autoclaved caging for the first month after irradiation. Mice were weighed at the same hour on the same 2 days of each week.

**In vivo LPS challenge.** Bone marrow chimeric mice that were age- and gender-matched were challenged in vivo with a sublethal dose of 12.5 mg kg$^{-1}$ UltraPure LPS (Invivogen) by intraperitoneal injection at an age of ~20 weeks as described[46]. Sample size was chosen based on the literature[47]. Mice were closely watched for up to 48 h and euthanized if moribund, which was considered a death under the terms of the experiment. Moribund status was determined by loss of righting reflex (LORR). At 48 h, all surviving mice were euthanized to minimize additional discomfort. This procedure was approved by our facility's Institutional Animal Care and Use Committee. A log-rank test (Mantel–Cox test) was used to compare the groups' survival distributions from initial challenge to time when LORR was observed. The Bonferroni correction was applied to the results for multiple curve analysis.

**ANA detection.** Blood was collected from 6- to 7-month-old female mice from all groups except the *Tnfaip3*–/– group (3–4 weeks) and allowed to clot for 1 h at room temperature. Serum was prepared by centrifugation at $15\ 871 \times g$ for 2 min. Supernatant was saved, aliquoted, and immediately stored at −80 °C prior to analysis. To detect ANA levels, samples were loaded onto a Hep-2 antigen substrate slide (MBL, Cat. # AN-1016) and incubated with a 1:800 dilution of Alex Fluor® 488 goat anti-mouse IgG (Jackson ImmunoResearch, Cat. # 115-546-003). Mounting medium (Fluoromount-G, Southern Biotechology, Cat. # 0100-01) plus Prolong Gold DAPI (Life Technologies, P-36931) was added, slides were sealed, and visualized with a Nikon Eclipse NI-E Fluorescence microscope (Nikon Instruments, Melville, NY, USA). Green fluorescence corresponds to IgG antibody location.

**Generation of DE4 and TT>A enhancer deletions in 293T cells.** High-scoring guide RNAs (gRNAs) with target sequences of 20 bp were selected using CRISPR Design web server by submitting the target sequence of the human *TNFAIP3* gene (http://crispr.mit.edu). Selected target sequences were cloned as complementary oligonucleotides into BsaI site of a pMB60 plasmid[48] (47941, Addgene) that contains a T7 promoter and the sgRNA backbone for the production of gRNA. The vector was digested with *Dra*I and gRNAs were transcribed in vitro using MAXIscript T7 and purified using the MEGAshortscript T7 Transcription kit (Life Technologies) and a MEGAclear Kit (Life Technologies). The RNA concentration was measured using a NanoDrop spectrophotometer and the quality was determined on a denaturing agarose gel. Two gRNAs targeting the ends of each enhancer were transfected into 293T cells stably expressing Cas9 (SCL-02-CA2, GeneCopoeia). Briefly, 1 day prior to transfection, 293T cells stably expressing Cas9 (293T-Cas9) were plated onto 24-well plates at $1.3 \times 10^5$ cells per well in Opti-MEM without antibiotics. On the day of transfection, 25 μl of Opti-MEM medium was added to a 1.5 ml sterile Eppendorf tube, followed by the addition of 160 ng gRNA. This solution was mixed by tapping briefly and then was incubated at 25 °C for 5 min. Meanwhile, 25 μl Opti-MEM medium was added to a separate sterile Eppendorf tube, followed by addition of 1.5 μl of Lipofectamine RNAiMAX. After briefly tapping, the Lipofectamine RNAiMAX solution was incubated at 25 °C for 5 min. After incubation, the gRNA solution was added to the Lipofectamine RNAiMAX solution. The sample was mixed and then incubated at 25 °C for 5 min and added dropwise onto 293T-Cas9 cells. At 48–72 h post transfection, the cells were trypsinized, single colonies were obtained by limiting dilution, and clones with deletion of enhancer at both alleles were identified by PCR. The PCR products were sequenced to confirm the deletion and the locus integrity (Supplementary Figs. 8a–c and 10f and Supplementary Table 6). The sizes of deletions at DE4 and TT>A are, respectively, 1489 and 789 bp.

**Generation of enhancer deletions in CTR mice using CRISPR.** We used in vitro fertilization to produce embryos from hA20(CTR) males and C57BL/6 females. hA20(CTR) males were used as sperm donors at 12 weeks of age and C57BL/6 females were used for oocytes at 8–10 weeks old. Swiss Webster mice were used as recipients of injected zygotes at 8–16 weeks of age. Pronuclear injections of Cas9 protein (50 ng μl$^{-1}$) with two gRNAs (50 ng μl$^{-1}$), selected on the basis of efficient deletion in 293T cells, were performed at the Rockefeller University transgenic core facility. Genomic DNA from founders was isolated from tail lysate by phenol-chloroform, and recovered by alcohol precipitation. To screen for deletion clones, primers flanking the outside of the CRISPR sgRNAs target sequences were designed and amplified with Taq polymerase. Given efficient CRISPR cutting and repair of DNA through non-homologous end joining, a ~370 bp product is expected for DE4 deletion and ~350 bp products is expected for TT>A deletion.

The PCR products from the positive founders were sequenced to confirm the deletion and the locus integrity after CRISPR/Cas9 manipulation (Supplementary Figs 8e, 9d and Supplementary Tables 5 and 6). The sizes of deletions at DE4 and TT>A are, respectively, 1474 and 795 bp. The off-target analysis was performed by PCR for the predicted off-target sites with the tail DNA samples and followed by T7E1 cleavage assay and Sanger sequencing. The off-target cleavage at alternate genomic sites was not detected in our analyses.

**Generation of mice with TT>A deletion using BAC modification.** The TT>A BAC clone was generated starting with hA20(CTR-BAC) using two-step BAC engineering approach. This system relies on a shuttle vector containing R6kγ-origin, an ampicillin-resistance gene for positive selection, a RecA gene, and a highly efficient negative selectable marker (sacB) to enhance the removal of unwanted vector sequences from the manipulated BAC in the presence of sucrose. The entire procedure involves the cloning of two homology arms with desired mutation and two recombination steps (Supplementary Fig. 9). Briefly, two recombination arms were amplified by PCR from the CTR-BAC DNA and sub-cloned into a targeting vector, pLD53AB. The modified targeting vector was transformed into BAC competent cells by electroporation and the transformed cells were selected with chloramphenicol and ampicillin to obtain co-integrants, which were confirmed by PCR and Southern blot analysis. The co-integrants were then selected with sucrose to eliminate the unwanted sequences and obtain the correctly modified BAC, which were determined by PCR and Southern blot. The PCR products from the desired mutant clones were sequenced to confirm the deletion, which encompassed 653 bp. BAC DNA preparation and pronucleus injection were performed as described above. BAC-mediated homologous recombination is able to delete exactly the sequence we want, whereas CRISPR-mediated deletion requires the PAM sequence on the region to be edited and NHEJ repair pathway that results in small nucleotide insertions or deletions (InDels) at the DSB site. The relative genomic locations of the CRISPR-mediated and recombineering-mediated deletions are shown in Supplementary Fig. 10e.

**Analysis of arthritis and dactylitis.** Mice were monitored weekly for the development of peripheral arthritis, which was scored by two observers (one blinded) based on more than two digits of the hind paws showing swelling and redness in repeated measurements. CTR, Δ-DOWN, and Δ-TT>A mice were sent to the Laboratory of Comparative Pathology at Sloan Kettering Institute for necropsy, histological analysis, and immunohistochemistry. Following euthanasia via $CO_2$ inhalation, tissues were harvested and fixed in 10% neutral buffered formalin. Following fixation, bones were demineralized using a formic acid-containing solution (Leica Decalcifier 1), and tissues were processed using an automated processor and routine tissue embedding in paraffin. Tissues were sectioned at 4 microns, and either stained with hematoxylin and eosin (H&E) or using the following antibodies: CD3 (Abcam ab16669, 1:100 following heat-induced epitope retrieval (HIER) in a pH 9.0 buffer); B220 (BD Pharmigen 550286, 1:200, HIER pH 6.0); Iba-1 (Abcam ab5076, 1:2500, HIER pH 9); myeloperoxidase (Dako A0398, 1:1000, HIER pH 6.0); and IgG (Vector BA-2000, 1:500, HIER pH 6). Staining was performed on a Leica Bond RX automated stainer using the Bond Polymer Refine detection system, Leica Biosystem DS9800.

Histology slides were reviewed by pathology staff at the Laboratory of Comparative Pathology and by a pathologist at HSS (Dr. Tania Pannellini) with training in musculoskeletal pathology in a blinded manner. Whole-mount sections of anterior and hind paws, including wrists and ankles, were stained with H&E and the synovium, peri-articular soft tissues, bone, and articular cartilage were reviewed for the extent of inflammatory infiltrates and tissue erosion. The morphologic parameters assessed to identify the presence of arthritis were as follows: synovial inflammation; synovial hypertrophy; synovial fibrosis; inflammation in articular cartilage; reactive and degenerative alterations in articular cartilage, like fibrocartilage formation or chondrocyte proliferation; inflammation in the subchondral bone; signs of bone resorption and deposition, such as scalloping of bone plates, proliferation of osteoclasts and osteoblasts; and inflammation or degeneration of articular and peri-articular soft tissues (entheses, tendons, muscles, and connective tissue). The presence of peri-articular dermatitis in the overlying skin was also evaluated and scored on the basis of morphological parameters: dermal edema and inflammation, epidermal acanthosis, epidermal exocytosis, and skin ulceration; and polymorphonuclear infiltrates.

**Flow cytometry.** Cell suspensions were prepared from total spleens. Red blood cell lysis was performed by incubation with Flow Cytometry Mouse Lyse Buffer (R&D Systems). Single-cell suspensions were then incubated with purified rat anti-mouse CD16/CD32 (mouse Fc block; BD Biosciences #553141 used at 1:50 dilution). Splenocytes were then incubated with anti-CD11b-APC (clone M1/70, BioLegend, 1:800 dilution) and anti- Ly-6G/Ly-6C (Gr1) (clone RB6-8C5, BioLegend, 1:300 dilution). Cells were analyzed on a FACS Canto (BD Biosciences) according the manufacturer's guidelines. Data were analyzed with FlowJo v10 software.

**FAIRE and ChIP-seq.** FAIRE experiments were performed as previously described[49]. Cells were treated with 1% formaldehyde for 7 min to crosslink chromatin. This was followed by the addition of 0.125 M glycine for 5 min to stop the crosslinking reaction. The cells were next washed with cold phosphate-buffered saline and scraped, followed by a second wash. Fixed cells were lysed in buffer LB1 (50 mM HEPES-KOH, pH 7.5, 140 mM NaCl, 1 mM EDTA, 10% glycerol, 0.5% NP-40, 0.25% Triton X-100, and protease inhibitors) for 10 min. Pelleted nuclei were resuspended in buffer LB2 (10 mM Tris-HCl, pH 8.0, 200 mM NaCl, 1 mM EDTA, 0.5 mM EGTA, and protease inhibitors) and incubated on a rotator for 10 min. The nuclei were pelleted and lysed in buffer LB3 (10 mM Tris-HCl, pH 8.0, 100 mM NaCl, 1 mM EDTA, 0.5 mM EGTA, 0.1% Na-deoxycholate, 0.5% N-lauroylsarcosine, and protease inhibitors). Chromatin was sheared using a Bioruptor Pico device (Diagenode). A total of 10% of sonicated nuclear lysates were saved as input. Phenol-chloroform-purified nuclear lysates and de-crosslinked input DNA were used for qPCR analysis using specific primers for the different enhancers analyzed (Supplementary Fig. 2 and Table 7). Chromatin accessibility is displayed relative to total input. ChIP-seq experiments for Pu.1 and C/EBP in LPS-stimulated monocytes were generated as part of the data set described in ref. [26] and were analyzed as previously described.

**Statistical analysis.** All data are expressed as means ± s.e.m. Statistical analyses were performed with GraphPad Prism (v. 5). The threshold of statistical significance was set at $P < 0.05$. Two-tailed Student's $t$-test was used to compare means in Figs. 2a, 5a, and Supplementary Fig. 13b–c. One-way analysis of variance ANOVA was performed to assess statistical significance for two or more groups followed by Tukey's post hoc multiple comparison test to detect intergroup differences in Figs. 2b, d, e, 3a, b, 4b, d, and 5d–f. Data in Fig. 3g were analyzed using paired $t$-test. Log-rank (Mantel–Cox) test was used to interpret statistical significance in Fig. 3d ($P < 0.0001$). No animals were excluded from the analysis. Experimental animals were grouped according to their genotypes and not randomized, except for Fig. 3a where animals were randomly selected from the colony for weighing and returned to their cages. The data displayed normal distribution. The estimated variance was similar between experimental groups.

**Data availability.** The data sets that support the findings of this study are available in the Gene Expression Omnibus database and assigned the accession number GSE104638.

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

## Acknowledgements

This work was supported by grants from the NIH (L.B.I.) and Lupus Research Institute 293032 (S.G.). The David Z. Rosensweig Genomics Center is supported by The Tow Foundation. The authors would like to thank Averil Ma for A20-deficient mice; Michel Nussenzweig for critically reading the manuscript and advice; Juana Gonzalez for assistance with transgene copy number determination and Kristen T. Ashourian for helping with generating the DE4 knockout mice; and Nathaniel Heintz, Feng Zhang, Cong Le, and Danwei Huangfu for their advice.

## Author contributions

L.B.I. and S.G. conceived and designed the experiments. U.K.S., M.P.L., L.F. and S.G. performed all experiments. U.K.S. and L.F. analyzed data and finalized the figures. S.P. and K.K. prepared Fig. 1. B.Z. provided technical advice. L.B.I., S.G., and M.P.L. wrote the manuscript with input from all authors. T.P. analyzed the pathology data. R.N. performed pronuclear injection. All authors read and approved the final manuscript.

## Additional information

**Competing interests:** The authors declare no competing financial interests.



