## [Peer Review File · Nature Communications]

Reviewers' comments:

Reviewer #1 (Remarks to the Author):

TNFAIP3/A20 is a critical negative regulator of inflammatory immune cell activation, human autoimmune responses and its gene locus is frequently mutated or silenced in human B cell lymphomas. Single nucleotide polymorphisms in the TNFAIP3 gene locus are significantly associated with a large number of human autoimmune diseases, such as systemic lupus erythematosus and rheumatoid arthritis. In their manuscript entitled "Dissection and Function of Autoimmunity-associated TNFAIP3 (A20) Gene Enhancers in Humanized Mouse Models" Liber and colleagues analyze the expression of A20 from different parts of its human gene locus transferred into mice via bacterial artificial chromosome (BAC) transgenesis. They thus demonstrate that the regulatory regions downstream of the A20 coding sequences are critical for basal and induced A20 expression in immune cells and synovial fibroblasts. They also show that loss of sequences 3' to the A20 coding sequences leads to autoantibody production in mice. This evidence seems to be based on 2 mice per genotype, which would not be sufficient. Furthermore, Liber and coworkers employ CRISPR/Cas9-mediated gene editing to remove a specific regulatory downstream region (DE4) from existing human A20 BAC transgenic mice. They also use BAC recombineering to produce BACs and BAC transgenic mice lacking the TTA enhancer. Their analyses suggest that DE4 has redundant functions at best. On the other hand, the TTA enhancer seems to be critical for basal and induced expression of A20 in lymphocytes. This last and maybe most relevant finding of the manuscript is unfortunately the least well described. It remains unclear how the enhancer was inactivated and how many different BAC transgenic founders were generated/analyzed.

Overall I find this manuscript highly relevant for the field. The authors generated state-of-the-art mouse models and present many novel results regarding the expression regulation of the human A20 gene locus in vivo. They demonstrate that tissue-specific enhancers critically control basal and induced gene expression and they provide (preliminary) evidence that loss of enhancer function can contribute to autoimmunity. All this is novel and important. However, the authors should rule out, as much as possible, that their results are biased by positional effects of BAC transgene integration, which can produce dramatically different outcomes.

Major concerns:

- 1) On page 6 line 134 the authors state that they "confirmed results using additional mouse lines". This would be very strong evidence that relevant statements are not or only minimally affected by position effects on the integrated BAC transgenes. However, I did find these confirmatory experiments in the manuscript.
- 2) Western Blots should be quantified throughout and means of all independent experiments shown.
- 3) How many mice were tested for the development of autoantibodies (Fig. 3e, Fig. S4)? If only two mice were tested per genotype this would not represent definitive proof in my mind.
- 4) The information regarding the TTA deletion mice is minimal. What region of the TTA enhancer is deleted/mutated? How many BAC transgenic mice were generated? What is their copy number? Were more than one TTA deletion BAC transgenic mouse strain analyzed to exclude effects of positional integration? In this regard the CRISPR/Cas9 approach chosen to delete DE4 in existing BAC-transgenic mice is superior.

Minor comments:

- 5) The authors should show the evidence that the complete BAC transgenes indeed integrated as shown in their schemes and depict the binding sites of the PCR primers used for verification of the various BAC transgenes in the supplement.
- 6) What is the size of the band detected by Southern Blot in Figure S1b? This corresponds to the region of the human BAC that is hereby proven to be integrated.
- 7) Quantification of the Western Blot shown in Figure S1e would be informative, to compare to the

gene expression data in S1d.

8) Basal resting expression of A20 seems to be significantly up in synovial fibroblasts from delta-up mice compared to CTR and delta-down mice (Figure 2C and Figure S2b). Is this a consistent finding? If yes, this should be mentioned and discussed.

9) Why did the authors analyze the protein levels of Cxcl10 (Fig. 2e)? What is special about the regulation of Cxcl10 by A20?

10) Figure S2C should depict hA20, not hA2.

11) 3f and S4b: what dynamic range is represented by the colors in the array results?

12) In Figure 4g the stimuli used with the respective cells should be shown. Western blots should be quantified.

13) PCR primers to verify the DE4 deletion mice are not given? Locus integrity after CRISPR/Cas9 manipulation (PCR assays should suffice)?

Reviewer #2 (Remarks to the Author):

In this manuscript, Gong et al. report a useful BAC-transgenic mouse in which the human TNFAIP3 locus compensates for A20 deficiency in the mouse. They use this model to map enhancers that control transgenic expression of the human TNFAIP3 gene. The authors use existing ENCODE epigenomic data sets to guide their search for enhancers in the human TNFAIP3 locus. This is fine, but they describe this approach as though they did the analyses (...we analyzed...we identified...). They simply viewed existing analyses in the UCSC browser - this needs to be clarified, and the majority of the visuals in figure 1 should be relegated to the supplementary materials. Also, the authors report no epigenomic mapping of the chromosome and chromatin structure of the BAC-encoded human TNFAIP3 locus inserted into the mouse genome. They cannot presume that these randomly inserted, transgenic constructs have the same epigenomic architecture as the locus has in its native chromosomal context in the human genome. CRISPR-based targeting of these enhancers in fully human B lymphoma and/or monocytic cell lines would also help to validate the authors' conclusions about the role of these regions in transcriptional control of TNFAIP3 expression. The results in figures 2 and 3 on effects of are convincing and valuable, although the authors should do some simple immunophenotyping of the mice to better understand the basis of the physiologic changes seen in the enhancer deleted line. The results for the TTA enhancer in figure 4 are convincing, but the results in support of a role for the DE4 region as an enhancer are not convincing. More data are needed to support their conclusions about the role of this region in TNFAIP3 expression.

Reviewer #3 (Remarks to the Author):

Review of the manuscript NCOMMS-17-06345 (Liber et al.)

The human TNFAIP3 (A20) gene has been associated with SLE through mapping of multiple disease-relevant SNPs, while a strong autoimmune phenotype was observed in Tnfaip3 null mice. Moreover, one polymorphic dinucleotide (TT>A) was shown to affect binding of the transcription factor NF- κ B and to be located in a downstream enhancer of TNFAIP3. However, this downstream enhancer was identified only by transient transfection experiments in established cell lines, which is a rather artifact-prone assay for enhancer detection. In this manuscript, Liber et al. have use a BAC transgenic approach to identify critical enhancers controlling TNFAIP3 in transgenic mice. They analyzed three different BAC transgenes, which differed in the extent of sequences located upstream and downstream of the TNFAIP3 gene. These BAC transgenes were crossed into Tnfaip3 null mice, and the basal or stimulated expression of the human TNFAIP3 gene was analyzed in splenic B and T cells as well as in bone marrow-derived macrophages (MBDM) and synovial fibroblasts. These experiments identified a downstream region of the TNFAIP3 gene, which is critical for basal and induced expression of TNFAIP3 and for the suppression of autoimmunity. This downstream region contains four DHS sites (DE1-DE4)

and the TT>A-containing region. CRISPR-Cas9-mediated deletion of DE4 and the TT>A region demonstrated a minimal effect of the DE4 deletion only in macrophages (see comment below) and a stronger effect of the deletion of the TT>A region exclusively in B and T cells. In summary, these transgenic in vivo data have unequivocally identified the TT>A-containing region as a potent enhancer in B and T lymphocytes, which is an important finding for the SLE field. Whether these data in the absence of a more detailed characterization of the TT>A enhancer are sufficient for publication in Nature Communications will require an editorial decision. However, I cannot recommend the manuscript in its current form for publication before the comments described below will be adequately addressed.

Specific comments:

1) General comment: The writing of the paper and the presentation of the figures could be significantly improved. It also appears that the statistical analysis of the presented data could be strengthened. For instance, it is mentioned on page 6 (line 134) that the results obtained with copy number-matched BAC lines were confirmed by analyzing additional BAC transgenic mouse lines. However, these data are not shown in supplementary figures except in Supplementary Figure 1d,e, although the data shown in panel d are not very convincing.

2) Figure 1. Misleading different alignment of the BACs in Figure 1a and 1c. hA20B(Δ UP-BAC) lacks the 5' end of the TNFAIP3 gene in panel a, whereas the entire gene is contained within this BAC in panel b (most likely correct).

3) Figure 1 contains largely reanalyzed data from other papers. I would recommend to minimize and to refocus the data of this figure by discussing the Hi-C data first (panel a and c together). The DNase-seq data (b and d) should be consolidated and focused only on the region shown in panel d, so that the DHS sites in the Δ UP and Δ DOWN regions are better presented and visible. Consolidation of the panels would avoid showing the same DNase- and ChIP-seq data as well as the BAC schemes multiple times.

4) Figure 2. The hA20(CTR) (gray) and hAD20(DOWN) (pink) BAC transgenes are considered to be expressed similarly under non-stimulated conditions (page 6-line 143), although they show significantly different expression in Figure 2a, which was neither statistically analyzed nor mentioned. The fact that the human A20 primers do not detect the mouse A20 transcript is excessively shown in the empty lanes 'C57BL6' and 'A20-/-' of the RT-PCR analysis in Figure 2b.

5) Figure 3. In the current version of Figure 3e, the ANA staining is very poorly visible. Is the ANA staining indeed so weak, which might indicate only a minor autoimmune phenotype. Furthermore, a statistical analysis of multiple mice is missing.

6) Figure 4. Panel a has not been polished as there is still some irritating junk for the browser view present (at left). Panel c is non-informative and could easily be deleted also because the same DHS track is already shown in panel a. Panel d; this reviewer is not convinced that there is a significant difference in the LPS-induced A20 expression in BMDMs between the hA20(CTR) and hA20(CTR- Δ DE4) mice. In other words, the deletion of DE4 seems to have no effect even in macrophages. Panel f; the significance assignment is irritating, as significance is indicated for the comparison of different time points (bracket), as it should be. However, stars or 'ns' are also shown for individual bars without indicating what comparison is meant in this case. Panel g; The immunoblot analysis (top row) should indicate that the A20 protein expression is reduced in stimulated B cells of hA20(CTR- Δ TTA) mice compared to hA20(CTR) mice. In most other immunoblot analyses, the upper non-specific band remains constant, whereas the low A20-specific band differs depending on the BAC transgene analyzed. In this case of Figure 4g, the unspecific band is also decreased so that the ratio of the unspecific to the specific band is the same for both the hA20(CTR) and hA20(CTR- Δ TTA) BAC transgenic mice analyzed. This should be clarified by demonstrating whether the same effect is seen in different independent immunoblot experiments.

7) Figure 4a. The DHS site DE1 should be indicated (most likely next DHS site upstream of DE2).

8) ANA staining with serum from hA20(CTR- Δ TTA) BAC transgenic mice. As the TT>A SNPs have been

associated with SLE, it would be important to demonstrate that the deletion of the TT>A enhancer in hA20(CTR- Δ TTA) mice results in an autoimmune phenotype. Hence, the corresponding ANA stainings should be shown, as they are so far missing.

9) Autoantibody detection. The data of the autoantibody detection experiments are not shown in quantitative way, as no scale bar describing the values corresponding to the different colors is shown (Figure 3f and Supplementary Figure 4b).

10) The some data are shown twice in the manuscript. 1) Only one set of ANA staining are shown, as the data of Figure 3a are the same as those shown in the middle vertical row of Supplementary Figure 4a. As a consequence, there is no statistics as the staining of only one mouse is shown. By the way, the DAPI staining in Supplementary Figure 4a is useless, as both the DAPI and ANA stainings are close to invisible in this figure. 2) The data of Figure 3f are shown again in the middle of Supplementary Figure 4b. 3) The data of Figure 2c (bottom row) are the same as those of Supplementary Figure 2b.

11) Description of the enhancer deletions Δ -DE4 and Δ -TTA. The deletion of the TT>A enhancer is not described at all, which is unacceptable. The deleted sequences should be clearly shown in a supplementary figure. Likewise, it would be advisable to show the sequences of the deleted DE4 enhancer. Currently only the sequences flanking the DE4 deletion are shown, and thus the interested reader has to retrieve these sequence from public databases.

12) Citation of TNFAIP3 SNPs. The papers describing the SLE-relevant TNFAIP3 SNPs (ref. 23, 24, 28) should be cited earlier in the manuscript, when the disease-relevant SNPs are first mentioned (pages 3-line 64).

13) Although superenhancers (mentioned in the abstract and result section) are very popular for marketing reason, it is a rather stupid concept as is best illustrate for TNFAIP3 gene. The entire domain (Figure 1b) containing DHS sites from far upstream regions to downstream regions including the gene itself is considered to be one superenhancer. I would recommend not using this non-informative term.

14) Supplementary Figure 2c. The figure legend mentions 'Immunoblot analysis of hA20 expression', although the anti-hA20 antibody detects the mA20 protein in mouse B cells (C57BL/6) and the hA20 protein in all other lanes of panel c. The description and labeling should be corrected accordingly.

Response to Reviewers

We thank the Reviewers for their time and encouraging and insightful comments. We are pleased that the Reviewers found the work of considerable potential interest: *“highly relevant for the field” “state-of-the-art mouse models” “many novel results” “novel and important” “results are...convincing and valuable” “unequivocally identified the TT>A region as an enhancer...important finding”*. We have experimentally addressed essentially all of the points raised by the Reviewers and have revised the manuscript accordingly. Changes in the manuscript have been highlighted.

Reviewer #1

Major points

1. *“On page 6 line 134 the authors state that they “confirmed results using additional mouse lines”. This would be very strong evidence that relevant statements are not or only minimally affected by position effects on the integrated BAC transgenes. However, I did find these confirmatory experiments in the manuscript.”*

Response: These experiments are now shown in Supplementary Figs. 2 and 3. Out of 12 lines originally generated with verified integration of intact BAC transgenes, 7 were further analyzed based on transgene copy number (the other lines had very high copy numbers). Most importantly for our manuscript, the results show copy-number dependent expression in three lines with the control CTR transgene and that both lines harboring the Δ -DOWN transgene showed low hA20 expression (also copy number dependent, Supplementary Fig. 3a). We appreciate the Reviewer's point about position effects and address it further in response to point 4 below.

2. *“Western Blots should be quantified throughout and means of all independent experiments shown.”*

Quantitation of blots is shown in Supplementary Fig. 11 and means of independent experiments are shown.

3. *“How many mice were tested for the development of autoantibodies (Fig. 3e, Fig. S4)? If only two mice were tested per genotype this would not represent definitive proof in my mind.”*

We thank the Reviewer for this point, which we followed up by sending an additional 20 samples, including 4 controls (non-transgenic mice matched for genetic background and age) to the same core facility for additional testing. The core facility seems to have changed personnel and procedures. Whereas our original samples (albeit only 2 per genotype) showed clean and internally consistent results, in the repeat experiments there were multiple false positives in the negative controls (including nontransgenic wild type mice). Overall the data were not internally consistent. Thus, although aspects of our previous data were reproduced (for example anti-histone antibodies) overall we do not feel comfortable showing these data and are not confident that any technical issues at the core will be sorted out in the foreseeable future. Our approach has been to remove these data. In contrast, the ANA data, which were obtained locally, were highly reproducible, have been quantitated, differences were statistically significant, and the

data are shown in Supplementary Fig. 7.

4. *“The information regarding the TTA deletion mice is minimal. What region of the TTA enhancer is deleted/mutated? How many BAC transgenic mice were generated? What is their copy number? Were more than one TTA deletion BAC transgenic mouse strain analyzed to exclude effects of positional integration? In this regard the CRISPR/Cas9 approach chosen to delete DE4 in existing BAC-transgenic mice is superior.”*

We have taken the Reviewer’s suggestion and developed an additional mouse line in which we used the CRISPR/Cas9 approach to delete the TT>A enhancer and confirm its function (Fig. 5d, pg. 10). This was done in the context of the control CTR BAC (i.e., deleting the TT>A enhancer in oocytes from CTR mice) and thus in the same genomic location. The details of both recombineering and CRISPR/Cas9 approaches to generate TT>A enhancer deletions are now shown in Supplementary Figs. 9 and 10, as are the relative locations and sizes of the deletions (sequences also provided in Supplementary Tables 5 and 6); copy number was matched at 1 for all mice studied and results shown. A detailed description of the DE4 deletion is provided in Supplementary Fig. 8.

Although the literature suggests that, in contrast to conventional transgenes, BAC transgene expression is typically insulated and generally minimally dependent on integration position effects (discussed on pp. 4 and 12 and reviewed in ref. 18, which also discusses the advantages of the ‘genomically humanized’ approach), we agree with the Reviewer that it is valuable to consider position effects, and thus have shown experimentally that the sub-TAD deletion (point 1) and the TT>A enhancer deletion (point 4) each had similar effects in different mouse lines, thus providing *“very strong evidence that relevant statements are not or only minimally affected by position effects”*. We think this point affects all transgenic experiments (even those that use integration into “landing pads”, as the integrated transgene can interact with endogenous regulatory elements at what are typically highly active gene loci), and have added text to the discussion mentioning position effects and how we have experimentally addressed this potential caveat (pg. 13).

Minor comments:

5. *“The authors should show the evidence that the complete BAC transgenes indeed integrated as shown in their schemes and depict the binding sites of the PCR primers used for verification of the various BAC transgenes in the supplement.”*

This information is now included in Supplementary Figs. 2, 8 and 10.

6. *“What is the size of the band detected by Southern Blot in Figure S1b? This corresponds to the region of the human BAC that is hereby proven to be integrated.”*

The size of the band (7.5 kb) is now included in the current Supplementary Fig. 2b.

7. *“Quantification of the Western Blot shown in Figure S1e would be informative, to compare to the gene expression data in S1d.”*

The quantitation of the blot is now included (currently Supplementary Figure 3d, e).

8. *“Basal resting expression of A20 seems to be significantly up in synovial fibroblasts from delta-up mice compared to CTR and delta-down mice (Figure 2C and Figure S2b). Is this a consistent finding? If yes, this should be mentioned and discussed.”*

This increase was consistent at both mRNA and protein levels (Fig. 2a, c and Supplementary Fig. 11a). Also, as indicated out in point 4 of Reviewer 3, basal expression was also elevated in BMDMs. We had missed this point, which is now discussed on pg. 7.

9. *“Why did the authors analyze the protein levels of Cxcl10 (Fig. 2e)? What is special about the regulation of Cxcl10 by A20?”*

CXCL10 is more highly expressed in serum than cytokines such as TNF or IL-6, thus facilitating its detection, and is elevated in multiple autoimmune diseases including RA and SLE. This is now described on pg. 7.

10. *“Figure S2C should depict hA20, not hA2.”*

This has been corrected (current Supplementary Fig. 5c).

11. *“3f and S4b: what dynamic range is represented by the colors in the array results?”*

This data has been omitted as discussed in response to point 3.

12. *“In Figure 4g the stimuli used with the respective cells should be shown. Western blots should be quantified.”*

The stimuli used have been added to the figure (currently Fig. 4d) and the quantitation of blots is shown in Supplementary Fig. 11.

13. *“PCR primers to verify the DE4 deletion mice are not given? Locus integrity after CRISPR/Cas9 manipulation (PCR assays should suffice)?”*

The PCR primer sequences are provided in Supplementary Table 4. Locus integrity around the DE4 deletion (and also around the newly generated CRISPR-mediated TT>A deletion) was assessed by PCR of the 5' and 3' ends and the hA20 gene body and further confirmed using Sanger sequencing, now shown in Supplementary Figs. 8 and 10.

Reviewer #2 (Remarks to the Author):

1. *“In this manuscript, Gong et al. report a useful BAC-transgenic mouse in which the human TNFAIP3 locus compensates for A20 deficiency in the mouse. They use this model to map enhancers that control transgenic expression of the human TNFAIP3 gene. The authors use existing ENCODE epigenomic data sets to guide their search for enhancers in the human TNFAIP3 locus. This is fine, but they describe this approach as though they did the analyses (...we analyzed...we identified...). They simply viewed existing analyses in the UCSC browser - this needs to be clarified, and the majority of the visuals in figure 1 should be relegated to the supplementary materials.”*

We did not mean to imply that we had carried out most of these experiments, and have revised the text (pp. 4-5) to make more clear when we used data sets generated by ENCODE, NIH Roadmap and other labs, and when we used data generated in our lab. We have also followed the suggestion of the Reviewer and moved 2 out of the 4 panels in this figure to Supplemental data. Although the data generated by others exists in public databases (also now clearly identified in the manuscript), and much of it can be easily visualized using the UCSC browser, our experience has been that substantial effort and time needs to be put into assembling the data at the locus of interest, understanding and visualizing the Hi-C data, and analyzing the data, i.e. bringing together the various types of data (Hi-C, DNase-seq, ATAC-seq, ChIPseq for histone marks, ChIP-seq for transcription factors) to understand chromatin structure, accessibility and regulation at the locus of interest, and then to interpret the BAC deletion results in this context. Thus, we believe that the use of “we analyzed” is appropriate, although we have toned down the descriptions overall in accord with the Reviewer’s comment. In addition, we clarify (pp. 4-5) that the ATC-seq, H3K27-Ac ChIP-seq, and PU.1 and C/EBP ChIPseq data shown in current Fig. 1b was generated in our lab. Although, to the Reviewer’s point, several of our data sets have been previously published and deposited into public data sets (as clarified in the text, pp. 4-5) these data still had to be analyzed at the A20 locus and integrated with the Hi-C data and BAC locations. Unpublished data on PU.1 and C/EBP ChIPseq under LPS-stimulated conditions has been newly deposited in GEO with accession number GSE104638.

2. *“Also, the authors report no epigenomic mapping of the chromosome and chromatin structure of the BAC-encoded human TNFAIP3 locus inserted into the mouse genome. They cannot presume that these randomly inserted, transgenic constructs have the same epigenomic architecture as the locus has in its native chromosomal context in the human genome.”*

We have used FAIRE assays, which in our system are specific and are very sensitive to changes in chromatin accessibility upon cell activation (for example, ref. 26), to confirm enhancer location and responsiveness to cell stimulation (increased chromatin accessibility) at the control CTR human A20 BAC in primary mouse BMDMs and B cells (Supplementary Fig. 4, described on pg. 6). These experiments also revealed that deletion of downstream enhancers, or of the TT>A enhancer, had minimal effects on basal or inducible chromatin accessibility at the hA20 promoter or other (non-deleted) enhancers.

3. *“CRISPR-based targeting of these enhancers in fully human B lymphoma and/or monocytic cell lines would also help to validate the authors’ conclusions about the role of these regions in transcriptional control of TNFAIP3 expression.”*

We appreciate the Reviewer’s point and worked very hard on obtaining homozygous enhancer deletions in four human cell lines. These experiments were technically successful in deleting both alleles of the TT>A enhancer in 293T cells, which resulted in decreased expression of hA20 (Fig. 5e, pg. 10 and Supplementary Fig. 10f), thus confirming the function of the TT>A enhancer in a fully human cell system.

Other than with 293T cells, in our experience it has been relatively straightforward to obtain cell lines deleted in one allele, but when these clones are expanded and the second allele is targeted, cells with homozygous deletions could be initially obtained, but upon cloning these cells uniformly died and homozygous null clones could not be grown out and analyzed.

Whereas B lymphoma cells have been shown to be dependent on A20 for survival, we were surprised by this finding in monocytic cell lines (THP-1 cells were exquisitely sensitive, and U937 cells somewhat more resistant but still died in culture after targeting the second enhancer allele). In our ongoing survey of the literature we were aware of two papers where CRISPR/Cas9 was used in THP-1 cells but closer examination suggests that stable cell lines were not maintained it was rare that 2 alleles were deleted. Additional review of the literature revealed that monocytes are impaired in DNA double strand break repair leading to excessive apoptosis (PNAS 2011, 108:21105 and PLoS One 2012, 7:e39956), and a Google forum discussion that CRISPR approaches can “trigger cell death” presumably in response to DNA in myeloid cells. Thus, we were not able to assess the effects of TT>A enhancer deletion in hematopoietic cell lines for technical reasons.

Without diminishing the importance of using CRISPR in cell lines to study enhancers, especially as this enables high through put screens and analysis of multiple enhancers, we hope that one can appreciate the importance of studying the function of enhancers in primary cells relevant for disease pathogenesis, and in systems that yield insights into (patho)physiological enhancer function in vivo (see also response to point 4 below). Thus, we have tried to present in our manuscript a balanced discussion of the strengths and weaknesses of various approaches, which acknowledges the caveats to our approach (pg. 13), but also points out cell lines do not always recapitulate patterns and mechanisms of gene expression or use the same enhancers as primary cells; this was actually one of the major criticisms of the early phases of the ENCODE project and helped motivate the move from cell lines to primary cells. More than 25 years of experience in one of the authors' labs (L.B.I.) has shown such substantial differences in magnitude and kinetics of gene expression between primary human monocytes and monocytic cell lines (or even complete lack of gene induction) that we have mostly abandoned cell lines for most analyses, as it seems likely that not all enhancers function in cell lines as they do in primary cells, and thus CRISPR experiments in these cell lines can lead to false negative results. Instead, to address the issue of testing enhancer function in fully human hematopoietic cells in future work, we are setting up a system to perform CRISPR/Cas9 experiments in iPSC, followed by differentiation into hematopoietic cell types, but this work is beyond the scope of this manuscript (and the October 10 resubmission deadline provided by the Editors).

4. *“The results in figures 2 and 3 on effects of are convincing and valuable, although the authors should do some simple immunophenotyping of the mice to better understand the basis of the physiologic changes seen in the enhancer deleted line.”*

We thank the reviewer for this point; since the original submission and during the revision process sufficient time has passed that we were able to analyze spontaneous development of phenotypes (which start at approximately 8 months of age) in TT>A enhancer deletion mice (new Figure 6, pg. 11), in addition to extending the phenotyping of the Δ -DOWN mice (4 enhancers deleted) (Fig. 3e-g, pg. 9). Surprisingly to us, the predominant disease manifestation spontaneously exhibited by both Δ -DOWN and Δ -TT>A mice was inflammatory arthritis and dactylitis (inflammation of digits) with synovitis in the hind paws. This phenotype mimics the phenotype of mice with myeloid-specific A20 deletion (refs. 34, 35), except it is more anatomically restricted (the myeloid A20-deleted mice also have involvement of ankle joints). Dactylitis is most commonly observed in autoimmune psoriatic arthritis, although it has also been observed in RA. In Δ -DOWN mice disease first became apparent at 6 months of age and reached an incidence of 84% by 12 months. In Δ -TT>A mice that preserve higher hA20

expression than Δ -DOWN mice, disease first became apparent at 8 months of age and reached an incidence of 60% after 12 months of age. In the context of the Δ -DOWN mice, these findings indicate that small decreases in A20 expression in several cell types (lymphocytes, myeloid cells, SFs) can phenocopy complete deletion in one cell type, namely myeloid cells. Most likely functional cooperation amongst several cell types with modest decreases in hA20 expression and thus increased cell activation potential results in a significant clinical phenotype in Δ -DOWN mice. Thus, broad defects in A20 expression in multiple cell types can result in a specific and focused clinical phenotype. Interestingly, the more modest effects of TT>A enhancer deletion on A20 expression, which were mostly restricted to lymphocytes, resulted in a similar disease phenotype, although with later onset and decreased incidence. The selective development of spontaneous arthritis in the paws and digits in our study may be related to long-term mechanical stress related to weight bearing, and to low A20 expression in the relevant tissue (stromal) cells, in this case synovial fibroblasts.

We think that these experiments provided one of the more interesting insights of our study. We believe that our system in which enhancer deletion modestly affects hA20 expression in several cell types more closely models the in vivo effects of subtle modulation of A20 expression by disease-associated noncoding SNVs in human patients than does complete A20 deficiency in one cell type in conditional knockout mice, and provides among the first insights into linkage of enhancers that control gene expression with specific disease phenotypes. Our system also sets the stage for future manipulation of mouse strain, genetic background, and environmental stressors and microbiota, to see how these influence disease phenotypes, possibly yielding insights into variable disease phenotypes associated with human A20 SNVs.

5. *“The results for the TTA enhancer in figure 4 are convincing, but the results in support of a role for the DE4 region as an enhancer are not convincing. More data are needed to support their conclusions about the role of this region in TNFAIP3 expression.”*

Based on this comment we performed an additional 3-4 experiments and also tested additional cell types (T cells, synovial fibroblasts), and found that the slight decreases in hA20 expression we previously observed were not statistically significant; thus these results are now described as negative data (pg. 10). We think it interesting (pg. 15) that this enhancer that had such a strong epigenomic ‘signature’ was redundant, whereas the TT>A enhancer that was much less apparent when chromatin was analyzed had a clear nonredundant function.

Reviewer #3 (Remarks to the Author):

Specific comments:

1. *“General comment: The writing of the paper and the presentation of the figures could be significantly improved. It also appears that the statistical analysis of the presented data could be strengthened. For instance, it is mentioned on page 6 (line 134) that the results obtained with copy number-matched BAC lines were confirmed by analyzing additional BAC transgenic mouse lines. However, these data are not shown in supplementary figures except in Supplementary Figure 1d,e, although the data shown in panel d are not very convincing.”*

We have edited the manuscript and the presentation of the figures according to the

suggestions of the Reviewers and for clarity. We now show additional supplementary data on the BAC deletion lines and have generated an additional TT>A deleted mouse line using CRISPR/Cas9 as discussed in detail above in response to points 2 and 4 of Reviewer #1.

2. *“Figure 1. Misleading different alignment of the BACs in Figure 1a and 1c. hA20B(Δ UP-BAC) lacks the 5' end of the TNFAIP3 gene in panel a, whereas the entire gene is contained within this BAC in panel b (most likely correct).”*

We have cleaned up Fig. 1a for clarity and marked the TNFAIP3 transcription start site with an arrow.

3. *“Figure 1 contains largely reanalyzed data from other papers. I would recommend to minimize and to refocus the data of this figure by discussing the Hi-C data first (panel a and c together). The DNase-seq data (b and d) should be consolidated and focused only on the region shown in panel d, so that the DHS sites in the Δ UP and Δ DOWN regions are better presented and visible. Consolidation of the panels would avoid showing the same DNase- and CHIP-seq data as well as the BAC schemes multiple times.”*

We have consolidated this figure, moved 2 panels to Supplementary data, and added ATACseq and CHIPseq data from our laboratory to former Fig. 1d (currently 1b); this is discussed in detail above in response to point 1 of Reviewer #2.

4. *“Figure 2. The hA20(CTR) (gray) and hAD20(UP) (pink) BAC transgenes are considered to be expressed similarly under non-stimulated conditions (page 6-line 143), although they show significantly different expression in Figure 2a, which was neither statistically analyzed nor mentioned. The fact that the human A20 primers do not detect the mouse A20 transcript is excessively shown in the empty lanes ‘C57BL6’ and ‘A20-/-’ of the RT-PCR analysis in Figure 2b.”*

The significant differences between the gray hA20(CTR) and the pink hA20(Δ -UP) cells are now marked in Fig. 2a and the consistent differences that were also apparent at the protein level (Fig. 2c) are discussed on pg.7. We have cleaned up the empty lanes in Fig. 2b for clarity.

5. *“Figure 3. In the current version of Figure 3e, the ANA staining is very poorly visible. Is the ANA staining indeed so weak, which might indicate only a minor autoimmune phenotype. Furthermore, a statistical analysis of multiple mice is missing.”*

The ANA staining was indeed weak relative to mice with full blown lupus (now clarified on pg. 8) but was reproducible and differences statistically significant, and is now shown in Supplementary Fig. 7b, c. Instead, in the main figures we focus on describing the striking spontaneous arthritic phenotype developed by both Δ -DOWN and Δ -TT>A mice, as described in detail in response to point 4 of Reviewer #2. Interestingly, the Δ -TT>A mice also showed a trend towards an elevated ANA by 11 months of age (Supplementary Fig. 13b, pg. 11), although this will require retesting at a much older age to determine statistical significance.

6. *“Figure 4. Panel a has not be polished as there is still some irritating junk for the browser view present (at left). Panel c is non-informative and could easily be deleted also because the same DHS track is already shown in panel a. Panel d; this reviewer is not convinced that there is a*

significant difference in the LPS-induced A20 expression in BMDMs between the hA20(CTR) and hA20(CTR-ΔDE4) mice. In other words, the deletion of DE4 seems to have no effect even in macrophages. Panel f; the significance assignment is irritating, as significance is indicated for the comparison of different time points (bracket), as it should be. However, stars or 'ns' are also shown for individual bars without indicating what comparison is meant in this case. Panel g; The immunoblot analysis (top row) should indicate that the A20 protein expression is reduced in stimulated B cells of hA20(CTR-ΔTTA) mice compared to hA20(CTR) mice. In most other immunoblot analyses, the upper non-specific band remains constant, whereas the low A20-specific band differs depending on the BAC transgene analyzed. In this case of Figure 4g, the unspecific band is also decreased so that the ratio of the unspecific to the specific band is the same for both the hA20(CTR) and hA20(CTR-ΔTTA) BAC transgenic mice analyzed. This should be clarified by demonstrating whether the same effect is seen in different independent immunoblot experiments.”

Figs. 4a and f have been cleaned up following the suggestions of the Reviewer. As discussed in response to point 5 of Reviewer #2, we performed additional experiments and decreases in hA20 expression in Δ-DE4 cells were not statistically significant; thus these results are described as negative data (pg. 10). We think it interesting (pg. 15) that this enhancer that had such a strong epigenomic ‘signature’ was redundant, whereas the TT>A enhancer that was much less ‘marked’ at the chromatin level had a clear function. Concerning panel g (currently panel 5c), we have reviewed all four experiments with B cells and now show a more representative blot where the upper nonspecific band does not change; quantitation of all 4 blots is shown in Supplementary Fig. 11c.

7. *“Figure 4a. The DHS site DE1 should be indicated (most likely next DHS site upstream of DE2).”*

DE1 is now indicated.

8. *“ANA staining with serum from hA20(CTR-ΔTTA) BAC transgenic mice. As the TT>A SNPs have been associated with SLE, it would be important to demonstrate that the deletion of the TT>A enhancer in hA20(CTR-ΔTTA) mice results in an autoimmune phenotype. Hence, the corresponding ANA stainings should be shown, as they are so far missing.”*

Deletion of the TT>A enhancer had a lesser effect on hA20 expression than did deletion of the downstream sub-TAD (Δ-DOWN mice). In accord with a delayed development and lower incidence of the arthritic phenotype, ANA staining in the Δ-TT>A mice only showed a trend towards an elevated ANA by 11 months of age (Supplementary Fig. 13b, pg. 11), although this will require retesting at a much older age to determine statistical significance. In addition to ANAs, in the revised manuscript we present the striking development of inflammatory arthritis in both lines.

9. *“Autoantibody detection. The data of the autoantibody detection experiments are not shown in quantitative way, as no scale bar describing the values corresponding to the different colors is shown (Figure 3f and Supplementary Figure 4b).”*

As discussed in response to point 3 of Reviewer #1, technical issues with the autoantibody testing core precluded strengthening these data and they have been removed.

10. *“The some data are shown twice in the manuscript. 1) Only one set of ANA staining are shown, as the data of Figure 3a are the same as those shown in the middle vertical row of Supplementary Figure 4a. As a consequence, there is no statistics as the staining of only one mouse is shown. By the way, the DAPI staining in Supplementary Figure 4a is useless, as both the DAPI and ANA stainings are close to invisible in this figure. 2) The data of Figure 3f are shown again in the middle of Supplementary Figure 4b. 3) The data of Figure 2c (bottom row) are the same as those of Supplementary Figure 2b.”*

In all three of these cases, the idea was, given space constraints, to show the key result in the main figure, and to show a fuller panel of data with additional controls in the Supplementary materials (in the case of blots to facilitate comparison of previous Supplementary Fig. 2a and 2b). In response to the Reviewer's comment, we now only show the data once. The ANA data are now shown in Supplementary Fig. 7b, c with quantitation and statistical analysis based on $n = 4$. The data in Fig. 3f (autoantibody staining) has been omitted as described above. Previous Supplementary Fig. 2b has been deleted.

11. *“Description of the enhancer deletions Δ -DE4 and Δ -TTA. The deletion of the TT>A enhancer is not described at all, which is unacceptable. The deleted sequences should be clearly shown in a supplementary figure. Likewise, it would be advisable to show the sequences of the deleted DE4 enhancer. Currently only the sequences flanking the DE4 deletion are shown, and thus the interested reader has to retrieve these sequence from public databases.”*

A detailed description of the enhancer deletions is now provided in Supplementary Figures 8-10, which includes a comparison of the deletions. The full sequences that were deleted, and those flanking the deletion, are shown in Supplementary Tables 5 and 6, respectively.

12. *“Citation of TNFAIP3 SNPs. The papers describing the SLE-relevant TNFAIP3 SNPs (ref. 23, 24, 28) should be cited earlier in the manuscript, when the disease-relevant SNPs are first mentioned (pages 3-line 64).”*

The manuscript has been modified accordingly.

13. *“Although superenhancers (mentioned in the abstract and result section) are very popular for marketing reason, it is a rather stupid concept as is best illustrate for TNFAIP3 gene. The entire domain (Figure 1b) containing DHS sites from far upstream regions to downstream regions including the gene itself is considered to be one superenhancer. I would recommend not using this non-informative term.”*

We have toned down the language about superenhancers.

14. *“Supplementary Figure 2c. The figure legend mentions ‘Immunoblot analysis of hA20 expression’, although the anti-hA20 antibody detects the mA20 protein in mouse B cells (C57BL/6) and the hA20 protein in all other lanes of panel c. The description and labeling should me corrected accordingly.”*

We have now clarified the legend and labeling (current Supplementary Fig. 5c).

REVIEWERS' COMMENTS:

Reviewer #1 (Remarks to the Author):

In their extensively revised manuscript Sokhi, Liber et al. have essentially addressed all my concerns. The paper is of very high significance for the field and should be published with high priority.

I have one remaining criticism: The situation with the detection of auto-antibodies is not ideal. The authors removed previously shown data on auto-antibodies as the core facility that generated them turned out to be unreliable. Therefore, ANA levels (supplementary figure 7B, C) are the only evidence for auto-antibody production and these data have to be better described and put into context: How many mice were analysed in each group (BL/6, CTR, delta-DOWN)? Individual measurements should be visible from the graph. How do the ANA levels in delta-DOWN mice compare to mice with SLE-like disease? These data should be included, as the authors state: "spontaneous development of weak (relative to mice with full-blown SLE) but statistically significant elevations in anti-nuclear antibodies (ANAs)".

Reviewer #2 (Remarks to the Author):

the authors have addressed the concerns to a satisfactory degree

Reviewer #3 (Remarks to the Author):

Review of the revised manuscript NCOMMS-17-06345A (Liber et al.)

The revised manuscript has been improved by cleaning up the data and their presentation as suggested by the first and third reviewers. Data that were no longer reproducible were eliminated, and the figures are now better presented. As requested, the authors now document how the TT>A enhancer of the human TNFAIP3 gene was deleted in the BAC- Δ TTA transgene. To my mind, the initially submitted manuscript should already have undergone the cleaning-up action that has now been preformed with the revised manuscript. The sequences indicating the deletion of the TT>A enhancer are shown in small and blurred letters (new Supplementary Figure 10b). Only by magnifying these sequences on the computer, I finally realized that the deletion of the TT>A enhancer is 655-bp long and thus quite large. Again, the authors have not mentioned the size of the deletion in the text of their manuscript. Instead, they mention in the discussion (page 13, 6th and 5th line from bottom) that "our work (i.e. deletion of the TT>A enhancer) sets the stage for future dissection of this enhancer, including using genome editing to introduce the TT>A risk allele and determine functional consequences in vivo". In a nutshell, this manuscript has functionally characterized the TT>A enhancer by demonstrating that its deletion reduces TNFAIP3 expression 2-3-fold in steady state and upon stimulation, which results in a mild autoimmune phenotype. For me, this result could have been the starting point for a more detailed analysis of the entire TT>A enhancer. As mentioned in my first review, it will require an editorial decision, whether these data in the absence of a more detailed characterization of the TT>A enhancer are sufficient for publication in Nature Communications.

Response to Reviewers

We are pleased that the Reviewers felt that the revised manuscript addressed their concerns, were enthusiastic about the manuscript "*The paper is of very high significance for the field and should be published with high priority*". We thank the Reviewers for their time and address their remaining minor comments below; changes in the manuscript are marked using track changes.

Reviewer #1

"The situation with the detection of auto-antibodies is not ideal...ANA levels (supplementary figure 7B, C) are the only evidence for auto-antibody production and these data have to be better described and put into context: How many mice were analysed in each group (BL/6, CTR, delta-DOWN)? Individual measurements should be visible from the graph. How do the ANA levels in delta-DOWN mice compare to mice with SLE-like disease? These data should be included, as the authors state: 'spontaneous development of weak (relative to mice with full-blown SLE) but statistically significant elevations in anti-nuclear antibodies (ANAs)'."

We now clarify in Supplementary Fig. 7 legend that there were 4 mice/group and show individual measurements and statistical analysis ($p < 0.01$, Supplementary Fig. 7c). The comparison of delta-DOWN mice and mice with SLE-like disease that show higher ANA levels is shown in new Supplementary Fig. 7d.

Reviewer #2 – no additional comments

Reviewer #3

"The sequences indicating the deletion of the TT>A enhancer are shown in small and blurred letters (new Supplementary Figure 10b). Only by magnifying these sequences on the computer, I finally realized that the deletion of the TT>A enhancer is 655-bp long and thus quite large. Again, the authors have not mentioned the size of the deletion in the text of their manuscript."

The lettering has been made more clear in Supplementary Fig. 10b and the size of the TT>A enhancer deletion is also clarified in the text on pg. 10.

"For me, this result could have been the starting point for a more detailed analysis of the entire TT>A enhancer. As mentioned in my first review, it will require an editorial decision, whether these data in the absence of a more detailed characterization of the TT>A enhancer are sufficient for publication in Nature Communications"

We agree that these are important experiments, but given multiple potential binding sequences in the enhancer, the need to generate multiple transgenic mice to study each enhancer modification in primary cells, and the need to use the much less efficient process of CRISPR-mediated homologous recombination to generate more precise mutations, these experiments will take several years to complete and are beyond the scope of this manuscript. To our knowledge there are very few publications

describing detailed characterization of mammalian enhancers and these have been performed mostly using cell lines. We believe the current manuscript represents a state-of-the-art approach that to our knowledge is the first characterization of human enhancers using an in vivo system and primary cells, and causal linkage of enhancers to a specific autoimmune/inflammatory phenotype.